# The ESCRT machinery regulates retromer-dependent transcytosis of septate junction components in *Drosophila*

**Hendrik Pannen, Tim Rapp, Thomas Klein***

Institute of Genetics, Heinrich-Heine-Universität Düsseldorf, Düsseldorf, Germany

**Abstract** Loss of ESCRT function in *Drosophila* imaginal discs is known to cause neoplastic overgrowth fueled by mis-regulation of signaling pathways. Its impact on junctional integrity, however, remains obscure. To dissect the events leading to neoplasia, we used transmission electron microscopy (TEM) on wing imaginal discs temporally depleted of the ESCRT-III core component Shrub. We find a specific requirement for Shrub in maintaining septate junction (SJ) integrity by transporting the claudin Megatrachea (Mega) to the SJ. In absence of Shrub function, Mega is lost from the SJ and becomes trapped on endosomes coated with the endosomal retrieval machinery retromer. We show that ESCRT function is required for apical localization and mobility of retromer positive carrier vesicles, which mediate the biosynthetic delivery of Mega to the SJ. Accordingly, loss of retromer function impairs the anterograde transport of several SJ core components, revealing a novel physiological role for this ancient endosomal agent.

## Introduction

Developmental and physiological functions of epithelia rely on a set of cellular junctions, linking cells within the tissue to a functional unit. While E-cadherin-based adherens junctions (AJs) provide adhesion and mechanical properties, formation of the paracellular diffusion barrier depends on tight junctions (TJs). Proteins of the conserved claudin family play a key role in establishing and regulating TJ permeability in the intercellular space by homo- and heterophilic interactions with Claudins of neighboring cells (*Günzel and Yu, 2013*). Arthropods, such as *Drosophila*, do not possess TJs but a functionally similar structure in ectoderm-derived epithelia termed pleated septate junction (pSJ, SJ hereafter), characterized by protein dense septa lining the intercellular space in electron micrographs (*Gilula et al., 1970*). Structure and function of *Drosophila* SJs depend on a convoluted multi-protein complex containing at least a dozen components. Three claudins, among them Megatrachea (Mega), have been shown to be required for SJ formation and barrier function in flies (*Behr et al., 2003*; *Nelson et al., 2010*; *Wu et al., 2004*). Besides claudins, several transmembrane proteins (TMPs) such as Neurexin-IV (NrxIV), Neuroglian (Nrg) or ATPα contribute to the formation of the stable SJ core complex, which is characterized by low mobility within the membrane (*Baumgartner et al., 1996*; *Genova and Fehon, 2003*; *Oshima and Fehon, 2011*). At the intracellular side of the junction, cytoplasmic proteins such as Coracle (Cora), Varicose (Vari), and Discs large (Dlg) associate with the transmembrane components, contributing to the formation of a stable fence-like scaffold (*Bachmann et al., 2008*; *Lamb et al., 1998*; *Laval et al., 2008*; *Woods and Bryant, 1991*; *Wu et al., 2007*). While junction formation during embryogenesis requires the SJ localized cytoplasmic protein Dlg, this basolateral cell polarity factor is not a structural part of the immobile junction core complex (*Oshima and Fehon, 2011*; *Woods et al., 1996*). This explains the functional separation of barrier formation and apicobasal polarity despite the close association of Dlg-complex components with the SJ. Albeit growing knowledge about the structural composition of SJs, the intracellular events required for assembly and maintenance of SJ complexes remain

**\*For correspondence:**
thomas.klein@hhu.de

**Competing interests:** The authors declare that no competing interests exist.

**eLife digest** Proteins are large molecules responsible for a variety of activities that cells needs to perform to survive; from respiration to copying DNA before cells divide. To perform these roles proteins need to be transported to the correct cell compartment, or to the cell membrane. This protein trafficking depends on the endosomal system, a set of membrane compartments that can travel within the cell and act as a protein sorting hub. This system needs its own proteins to work properly. In particular, there are two sets of proteins that are crucial for the endosomal systems activity: a group of proteins known as the ESCRT (endosomal sorting complex required for transport) machinery and a complex called retromer. The retromer complex regulates recycling of receptor proteins so they can be reused, while the ESCRT machinery mediates degradation of proteins that the cell does not require anymore. In the epithelia of fruit fly larvae – the tissues that form layers of cells, usually covering an organ but also making structures like wings – defects in ESCRT activity lead to a loss of tissue integrity.

This loss of tissue integrity suggests that the endosomal system might be involved in transporting proteins that form cellular junctions, the multiprotein complexes that establish contacts between cells or between a cell and the extracellular space. In arthropods such as the fruit fly, the adherens junction and the septate junction are two types of cellular junctions important for the integrity of epithelia integrity. Adherens junctions allow cells to adhere to each other, while septate junctions stop nutrient molecules, ions and water from leaking into the tissue. The role of the endosomal system in trafficking the proteins that form septate junctions remains a mystery.

To better understand the role of the endosomal system in regulating cell junctions and tissue integrity, Pannen et al. blocked the activity of either the ESCRT or retromer in wing imaginal discs – the future wings – of fruit fly larvae. Pannen et al. then analyzed the effects of these endosomal defects on cellular junctions using an imaging technique called transmission electron microscopy. The results showed that both ESCRT and retromer activities are necessary for the correct delivery of septate junction components to the cell membrane. However, neither retromer nor ESCRT were required for the delivery of adherens junction proteins.

These findings shed light on how retromer and the ESCRT machinery are involved in the epithelial tissue integrity of fruit fly larvae through their effects on cell junctions. Humans have their own versions of the ESCRT, retromer, and cell junction proteins, all of which are very similar to their fly counterparts. Since defects in the human versions of these proteins have been associated with a variety of diseases, from infections to cancer, these results may have implications for research into treating those diseases.

largely unknown. Specifically, how proliferative tissues, such as the imaginal disc epithelium, maintain SJ integrity is not well established.

It was recently shown that newly synthesized SJ components integrate into the junction from the apical side (in between AJ and SJ) in a 'conveyor belt-like' fashion (*Babatz et al., 2018*; *Daniel et al., 2018*). In addition, SJ components are frequently associated with endosomal compartments, suggesting a role for the endosomal system in coordinating transport and turnover of SJ complexes (*Nilton et al., 2010*; *Tempesta et al., 2017*; *Tiklová et al., 2010*). Consistently, endocytosis is required to concentrate SJ components at the junctional region during embryogenesis (*Tiklová et al., 2010*). This suggests that passage of SJ TMP components (or the whole SJ protein complex) through the endosomal system may be a requirement for SJ formation, with the underlying mechanisms remaining poorly characterized.

The endosomal system fulfils a plethora of physiological functions by tightly regulating the intracellular transport of TMPs and membranes within the cell. Following endocytosis from the plasma membrane, TMPs enter the endosomal system where they undergo cargo specific sorting. This process provides separation of proteins destined for degradation from those that exit the endosomal system to be recycled. Two evolutionary conserved endosomal sorting machineries, the endosomal sorting complex required for transport (ESCRT) and the retromer complex, mediate cargo sorting into the degradative and recycling pathway, respectively (*Cullen and Steinberg, 2018*). To coordinate these opposing transport activities, the endosomal system comprises a highly dynamic

membrane network governing retromer-dependent tubulation for recycling and ESCRT-mediated generation of intraluminal vesicles (ILV) for degradation.

Endocytosed proteins can evade ESCRT-dependent packaging into ILVs by exiting the maturing endosome (ME) through tubular retrieval domains induced by specialized recycling machineries such as retromer. Initially characterized as a regulator of endosome-to-Golgi cargo retrieval in yeast, this endosomal agent comprises two subcomplexes that cooperatively drive cargo sorting into tubular recycling carriers (*Carlton et al., 2004*; *Horazdovsky et al., 1997*; *Seaman et al., 1997*). Similar to the ESCRT machinery, cargo clustering and membrane deformation is performed by distinct functional units within the retromer pathway. Motif-based cargo recognition and aggregation is mediated by the endosomally localized Vps26:Vps29:Vps35 complex, which has been termed cargo-selective complex (CSC) (*Lucas et al., 2016*; *Nothwehr et al., 2000*; *Seaman et al., 1997*). Since the ancient CSC does not possess membrane bending activity, cooperation with tubulating factors such as proteins of the SNX-BAR (Sorting Nexin-Bin/Amphiphysin/Rvs) family is required for recycling carrier generation (*Cullen, 2008*). Proteins containing the curved BAR-domain can assemble into regular helical coats on endosomes, thereby inducing cytoplasm faced tubulation (*Frost et al., 2009*). Concerted action of CSC stably complexed with SNX-BAR proteins to retrieve endosomal cargo was initially characterized as the classical retromer pathway in yeast (*Horazdovsky et al., 1997*; *Seaman et al., 1998*). In metazoans however, retromer function is not restricted to SNX-BAR-dependent pathways. Specifically, cooperations of CSC with SNX3 or SNX27 (both lacking BAR-domains) emerged as alternative routes for endosomal retrieval (*Harterink et al., 2011*; *Lauffer et al., 2010*; *Steinberg et al., 2013*). Proteomic data from mammalian cells suggest that surface levels of well over 100 TMPs depend on retromer and many of these proteins seemingly interact with CSC or SNX27 (*Steinberg et al., 2013*). Recently, *Drosophila* has proven invaluable for assessing and confirming the physiological relevance of some of these putative retromer cargos in vivo (*Strutt et al., 2019*).

Cargo proteins within the endosomal system that do not undergo recycling can enter the degradative trafficking route starting with their sorting into ILVs. Generation of ILVs at the limiting membrane of MEs requires the canonical ESCRT function, which is performed by four in sequence acting complexes (ESCRT-0, -I, -II, III) and the ATPase Vps4 (*Babst et al., 2002b*; *Babst et al., 2002a*; *Babst et al., 1998*; *Bilodeau et al., 2002*; *Katzmann et al., 2001*). Ubiquitination of TMPs serves as the primary degradative sorting signal and sequestration of TMPs into ILVs is an essential prerequisite to complete lysosomal degradation. Several ESCRT components such as Vps27/Hrs (ESCRT-0) and Vps23/TSG101 (ESCRT-I) possess ubiquitin interacting motifs, which allow them to bind and cluster ubiquitinated TMPs (*Bilodeau et al., 2002*; *Katzmann et al., 2001*; *Shih et al., 2002*). Consequently, local concentration of ubiquitinated cargo by ESCRT complexes establishes a degradative subdomain at the endosomal membrane that is spatially separated from the retrieval subdomain (*Norris et al., 2017*; *Raiborg et al., 2002*; *Raiborg et al., 2001*). While ESCRT-0-II complexes provide cargo recognition and clustering, the membrane-deforming activity required to bud and abscise ILVs into the endosomal lumen depends on ESCRT-III components, which polymerize into helical arrays at the endosomal membrane (*Hanson et al., 2008*; *Saksena et al., 2009*). The most abundant ESCRT-III component is the highly conserved yeast Snf7/Vps32, encoded by the gene *shrub* (*shrb*) in *Drosophila* (*Sweeney et al., 2006*; *Teis et al., 2008*). Unlike upstream ESCRT components, ESCRT-III proteins only transiently assemble into a heterooligomeric complex at the endosomal membrane (*Teis et al., 2008*; *Wollert et al., 2009*). In consequence of ESCRT activity, the maturing endosome accumulates cargo-containing ILVs and is recognized in electron micrographs as a multi-vesicular body (MVB). The ESCRT/MVB pathway ends with Vps4-dependent dissociation of ESCRT-III components from the endosomal membrane. This step is required for the release of the nascent ILV and subsequent rounds of ILV formation (*Babst et al., 1998*; *Wollert et al., 2009*). Loss of ESCRT function was initially studied in yeast cells in which it led to the emergence of an aberrant pre-vacuolar endosomal organelle, termed class E compartment (*Raymond et al., 1992*). This defective endosomal structure is characterized by accumulation of degradative cargo and a failure to fuse with the vacuole/lysosome (*Doyotte et al., 2005*; *Raymond et al., 1992*).

The physiological relevance of ESCRT-mediated degradative TMP trafficking is particularly evident in *Drosophila* imaginal disc tissue. Here, loss of ESCRT function induces severe overgrowth, multilayering, apoptosis, and invasive behavior of the tissue; a phenotype attributed to mis-regulation of cellular signaling pathways, such as the Jak/Stat-, Jun-Kinase-, and Notch pathways

(*Herz et al., 2006*; *Moberg et al., 2005*; *Thompson et al., 2005*; *Vaccari and Bilder, 2005*). Consequently, ESCRT components were classified as endocytic neoplastic tumor suppressor genes (nTSG) in *Drosophila* (*Hariharan and Bilder, 2006*). While induction of over-proliferation and apoptosis in nTSG mutants have been extensively characterized, the events leading to loss of cell polarity and ultimately neoplastic transformation of the tissue remain poorly understood.

Here, we have analyzed the integrity of cellular junctions in an ESCRT-depleted wing imaginal disc epithelium to gain insight into the initial events leading to neoplastic transformation. To our surprise, preceding neoplastic overgrowth, we found a strong and specific reduction in the density of SJ. We show that ESCRT and retromer functions are required for anterograde transport of SJ components. By dissecting the intracellular trafficking itinerary of the claudin Megatrachea, we reveal that biosynthetic delivery of this core SJ component depends on a complex basal to apical transcytosis route relying on ESCRT and retromer functions.

## Results

### ESCRT knockdown specifically affects SJ integrity

To analyze the impact of ESCRT loss of function on junctional integrity, transmission electron microscopy (TEM) was used on wing imaginal discs that have been depleted of ESCRT function. We devised an RNAi-based approach allowing spatiotemporal knockdown of the ESCRT-III component Shrub. By using *hh*Gal4 and the temperature sensitive Gal4 Repressor (*tub*Gal80$^{ts}$), we specifically inhibited Shrub protein expression in the posterior compartment by expressing a UAS-*shrub*-RNAi construct (*Sweeney et al., 2006*) for specified durations. After 32 hr of RNAi expression, Shrub protein was effectively reduced in the posterior compartment as visualized by antibody staining (*Figure 1A*). ESCRT loss of function is known to induce high levels of apoptosis in imaginal disc tissue (*Herz et al., 2006*; *Thompson et al., 2005*; *Vaccari and Bilder, 2005*). Since apoptotic cells disassemble their junctions prior to extrusion from the tissue (*Brancolini et al., 1997*; *Steinhusen et al., 2001*), we co-expressed the viral caspase inhibitor p35 (*Hay et al., 1994*) with *shrub*-RNAi to preserve tissue integrity and allow unambiguous analysis of junctional structures. Interestingly, while 48 hr expression of *p35 + shrub*-RNAi yielded discs with no apparent morphological defects seen in a thin section (*Figure 1B*), discs after 65 hr of expression were disorganized and multi-layered in the posterior compartment as reported for *shrub* null mutant clonal eye disc tissue (*Figure 1C*; *Vaccari et al., 2009*). This indicates that our assay is able to reproduce the hallmarks of ESCRT-mediated neoplasia.

We used the 48-hr stage to analyze junctions by TEM prior to neoplasia. In the wildtype anterior control compartment, the membrane basal to the AJ was lined with the ladder-like electron-dense structures that represent the SJ (*Figure 1E*). In the posterior (*shrub*-RNAi) compartment however, only few electron-dense structures basally to the AJ were detected and an obvious ladder pattern was rarely seen (*Figure 1E′*). Interestingly, we did not find AJ appearance to differ noticeably between wildtype and *shrub*-RNAi compartments (*Figure 1E/E′*). We sought to quantify SJ integrity by measuring the total length of electron-dense structures basally to the AJ in a region of interest (ROI) with specified length of 2 µm. In anterior control compartments, roughly 32% of membrane within ROIs was lined with SJ (*Figure 1D*). This value decreased to about 15% in posterior compartments, indicating that 48 hr expression of *p35 + shrub*-RNAi reduces SJ density in wing discs by more than 50% (*Figure 1D*). Due to the lack of apparent neoplastic transformation at the 48 hr stage, we conclude that reduction in SJ density does not reflect indirect effects resulting from ESCRT induced epithelial to mesenchymal transition (EMT), but rather suggests a direct involvement of ESCRT in maintaining SJ density in wing imaginal discs.

We turned to fluorescence microscopy to analyze the subcellular localization of junctional proteins upon Shrub depletion. Again, we used depletion for 48 hr, which does not show apparent neoplastic overgrowth and thus preserves epithelial monolayer organization. We used antibodies against E-cadherin and the claudin Mega to reveal AJ and SJ, respectively. Both proteins localized almost exclusively at the apical membrane in wildtype cells of the anterior compartment of the disc (*Figure 1F*, left arrowheads). However, in the Shrub-depleted posterior compartment, Mega was not detected at the SJs, while E-cad showed wildtype AJ localization (*Figure 1F*, right arrowheads). The absence of Mega from SJs is also visible in maximum intensity projections of the junctional region

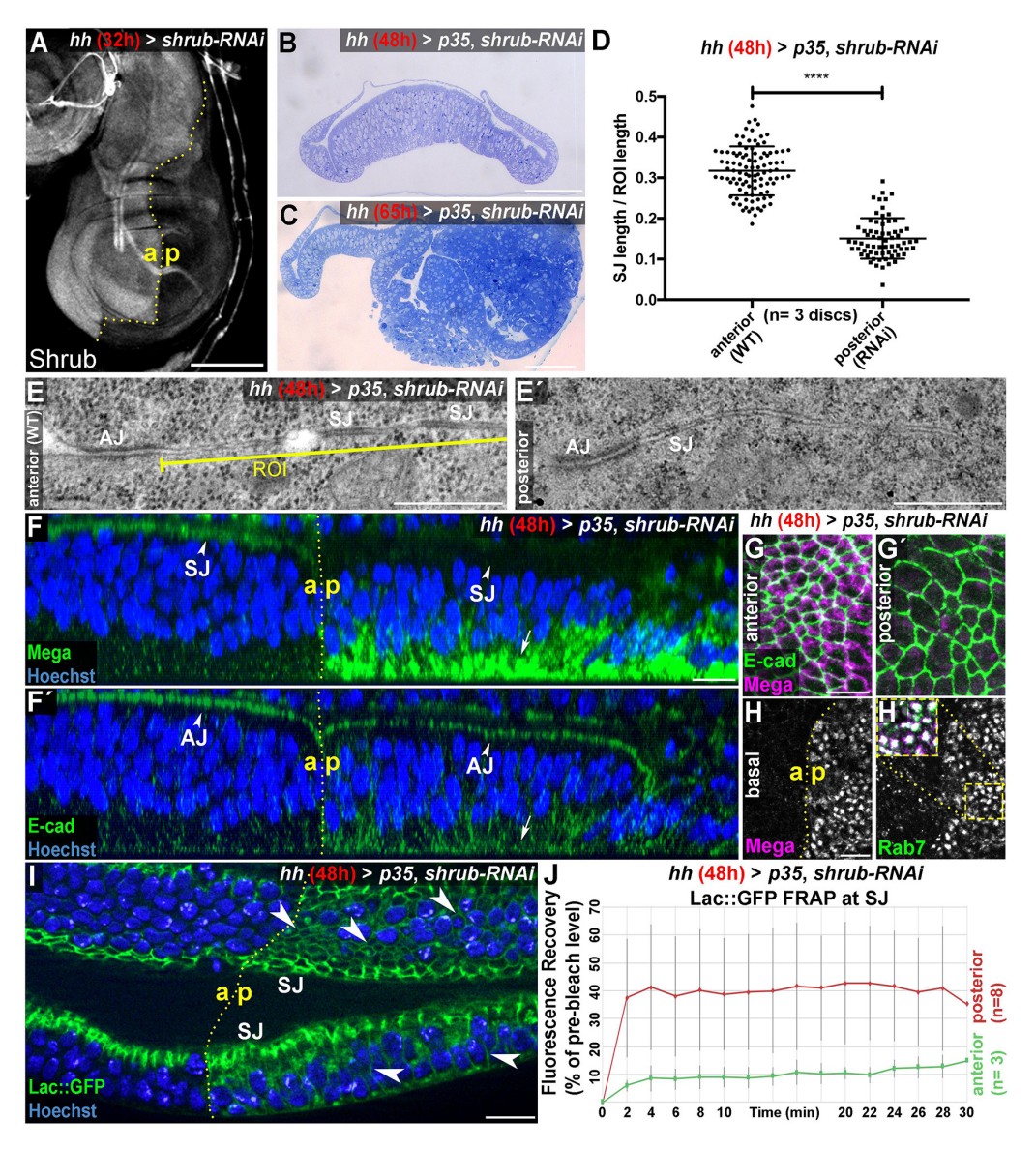

**Figure 1.** The ESCRT-III component Shrub is required for SJ integrity. (**A**) 32-hr expression of *shrub-RNAi* under the control of *hhGal4* effectively depletes Shrub levels in the posterior compartment of wing imaginal discs. (**B–C**) Thin sections of wing discs co-expressing *shrub-RNAi* and *p35* for either 48 hr (**B**) or 65 hr (**C**). Note the intact epithelial monolayer at the 48 hr stage (**B**) vs. posterior neoplastic overgrowth after 65 hr (**C**). (**D**) Quantification of electron-dense SJ in TEM sections reveals a ~50% reduction of SJ strands within Shrub-depleted compartments (48 hr) compared to control. Each data point represents a single junctional region of interest (ROI) of 2 μm length as defined in (**E**) A two-tailed t-test was used for statistical analysis with the significance level **** representing a p-value<0.0001. Representative ROIs of control (**E**) and Shrub-depleted compartments (**E'**) visualize the reduction in electron-dense SJ. (**F**) Optical section of a wing disc after 48 hr of Shrub depletion in the posterior compartment. The claudin Mega is lost from the apical SJ (compare arrowheads) and accumulates basal to the nuclei (arrow). In contrast, E-cad localization to AJ between the anterior control and the posterior compartment is largely unaffected (compare arrowheads in **F'**) while only little accumulation is seen at the basal pole (arrow). (**G**) Projections of the junctional area show the close association of E-cad with Mega in anterior control compartments (**G**) and apparent lack of junctional Mega in the Shrub-depleted tissue (**G'**). In basal planes, intracellular accumulation of Mega within the *shrub-RNAi* compartment is evident as a punctate pattern, colocalizing with the endosomal marker Rab7 (**H**). Subcellular localization of the SJ component Lachesin::GFP is restricted to SJ in anterior control compartments while showing extensive spread along the entire lateral membrane in Shrub-depleted cells (arrowheads in I). (**J**) Fluorescence recovery after photobleaching (FRAP) analysis of Lac::GFP at the SJ level reveals increased mobility in *shrub-RNAi* tissue, indicating defective fence function of SJ. a: anterior/p: posterior. The red graph displays mean Lac::GFP fluorescence recovery in the posterior compartment, the green graph represents mean values for the anterior control compartment. Error bars indicate standard deviation for each individual time point. Scale bar in (**A**) 100 μm, (**B–C**) 50 μm, (**E**) 0.5 μm, and all other panels 10 μm.

The online version of this article includes the following source data and figure supplement(s) for figure 1:

*Figure 1 continued on next page*

*Figure 1 continued*

**Source data 1.** Quantification of SJ density in WT and Shrub-depleted (+p35) tissue.
**Source data 2.** FRAP kinetics of Lac-GFP in WT and Shrub-depleted (+p35) tissue.
**Figure supplement 1.** Epithelial barrier function is compromised in Shrub or Vps26-depleted wing disc tissue.
**Figure supplement 2.** Mega depletion causes slight lateral spreading of Lachesin::GFP.
**Figure supplement 3.** Junctional levels of the core SJ component ATPα are reduced in Shrub-depleted wing imaginal disc tissue.

(*Figure 1G*). Importantly, Mega accumulated in vesicle like structures at the basal side of the cells, suggesting that it might be trapped in intracellular compartments (*Figure 1F*, arrow). This basal fraction of Mega colocalized with the endosomal marker Rab7, suggesting that Mega was trapped within maturing endosomes (inset in *Figure 1H'*).

We reasoned that junctional loss of Mega together with the decreased SJ density revealed by TEM analysis point toward defects in SJ integrity/function in *shrub*-RNAi tissue. Indeed, dye exclusion experiments suggest that epithelial barrier function is compromised in *shrub*-RNAi expressing wing discs (*Figure 1—figure supplement 1*). Results from another experiment also support this notion: The GPI-anchored SJ protein Lachesin (Lac) is required for barrier function and localized at the outer membrane leaflet (*Llimargas et al., 2004*). We visualized endogenous Lac localization by using a GFP protein trap line in *p35 + shrub*-RNAi expressing discs. While Lac::GFP was restricted to SJ in the anterior control compartment, the protein spread more laterally in *shrub*-RNAi tissue (*Figure 1I*, arrowheads).

Interestingly, this phenotype resembles mis-localization of SJ proteins in mutants of individual complex components during junction formation in the embryo. For example, in *mega* deficient embryos, NrxIV fails to concentrate at the SJ and localizes along the entire lateral membrane (*Behr et al., 2003*). We conclude that Shrub function maintains integrity of the SJ complex required for containment of Lac within the junction of the wing disc epithelium. Strikingly, when we depleted posterior wing disc compartments of Mega, we could only detect a faint lateral spreading of Lac::GFP basal to the SJ (*Figure 1—figure supplement 2*). This suggested that in wing discs, Shrub depletion more severely affects integrity of the SJ complex with respect to Lac::GFP confinement to the junction compared to Mega depletion. This made us wonder whether junction levels of other SJ core components might be affected upon Shrub depletion. Indeed, when we analyzed subcellular localization of the SJ core component ATPα upon 48 hr of *shrub*-RNAi expression, we found a strong and specific reduction of its junctional level in the Shrub-depleted compartment (*Figure 1—figure supplement 3*). This result indicates that the function of Shrub in regulating junctional membrane levels is not limited to Mega.

The lateral spreading of Lac::GFP within the membrane in Shrub-depleted tissue also suggested that its mobility might be increased. We measured fluorescence recovery after photobleaching (FRAP) of Lac::GFP to test this hypothesis. Consistent with a very stable and immobile SJ complex (*Oshima and Fehon, 2011*), Lac::GFP fluorescence recovery at the junction was low in the anterior control compartment, with GFP signal intensities barely reaching 20% of the pre-bleach levels half an hour after photobleaching (*Figure 1J*, green graph). In the *shrub*-RNAi compartment however, Lac::GFP fluorescence recovery at the SJ was quick and reached a plateau at roughly 40% of pre-bleach levels after a few minutes. This suggests that a fraction of Lac::GFP molecules shows increased mobility within the SJ membrane region in *shrub*-RNAi expressing cells, in line with a defective barrier/fence function of the SJ complex.

Altogether, these results indicate that SJ integrity critically depends on ESCRT function and suggest that Shrub is required for intracellular transport of Mega from an endosomal compartment toward the junction.

## The retromer CSC regulates membrane levels of SJ core components

The loss of Mega from the SJ and concomitant accumulation in basal aggregates cannot easily be explained with a role of Shrub in Mega degradative trafficking but rather suggest that its export from the endosomal system might be impaired in Shrub-depleted tissue. Consistent with a non-degradative ESCRT role in transport of Mega, we did not find Mega to accumulate intracellularly when we interfered with endosomal maturation or prevented endolysosomal fusion downstream of ESCRT

function, in contrast to the canonical ESCRT cargo Notch (*Figure 2—figure supplement 1*). These results suggest that Mega undergoes very little (if any) lysosomal turnover in wing imaginal discs and point toward an ESCRT function that is distinct from the degradative MVB pathway in trafficking of Mega.

We hypothesized that biosynthetic delivery of Mega to the SJ might require an endosomal recycling pathway depending on ESCRT for its proper function. To test this idea, we depleted imaginal discs of proteins known to regulate endosomal recycling and analyzed the impact on junctional Mega levels. We found that expression of an RNAi construct targeting the retromer CSC component Vps35 led to a significant reduction of Mega at the SJ (*Figure 2—figure supplement 1*), which led us to investigate the function of retromer in transport of SJ components. We generated null mutant clones of the retromer core component *Vps35* (*Vps35$^{MH20}$*, *Franch-Marro et al., 2008*) in wing discs and analyzed the subcellular distribution of junctional proteins. While Mega membrane levels at the SJ were reduced in *Vps35$^{MH20}$* tissue, junctional E-cad levels were unaffected (*Figure 2A*). This result is analogous to the phenotype seen upon *shrub*-RNAi expression (*Figure 1*).

We also found reduced membrane levels of the SJ core components ATPα, NrxIV and Nrg in *Vps35$^{MH20}$* clones, visualized by using endogenously tagged GFP protein trap lines (*Figure 2B–D*). These data suggest that retromer has a general function in SJ protein trafficking that is not restricted to Mega. In line with this, quantification of junctional signal revealed that all of the affected SJ core components show similarly reduced levels in *Vps35$^{MH20}$* tissue (roughly 50% compared to wildtype, *Figure 2H*). Additionally, we found *Vps35* to be required for regulating membrane levels of SJ components in several tissues, such as eye/leg imaginal discs and pupal wings, pointing to a common requirement for the retromer pathway in SJ protein transport (*Figure 2—figure supplement 2*). Surprisingly, loss of NrxIV::GFP from the SJ membrane in *Vps35$^{MH20}$* mutant pupal wing tissue was more severe compared to the phenotype seen in third instar wing discs, suggesting that SJs within a tissue that undergoes morphogenetic changes and does not strongly proliferate might be even more sensitive toward loss of retromer function (*Figure 2—figure supplement 2*).

Clones mutant for the Vps35 interaction partner *Vps26* phenocopied *Vps35$^{MH20}$* tissue with regard to Mega membrane levels, supporting a general function of the CSC in transport of SJ components (*Figure 2E*). Interestingly, in contrast to these structural SJ core components, the levels of the associated cytoplasmic scaffolding proteins Dlg and Cora were unaffected in *Vps35$^{MH20}$* mutant tissues (*Figure 2F,G*). This is consistent with *Vps35$^{MH20}$* clones maintaining intact apicobasal polarity and supports a role of retromer in trafficking specific structural components of SJs. Nevertheless, our extended analysis with further SJ components revealed that the GPI-anchored proteins Lachesin, Contactin, as well as the cytoplasmic Varicose, show reduced SJ levels in *Vps35$^{MH20}$* clones (*Figure 2—figure supplement 3*). This indicates that the retromer CSC regulates apical levels of a large subset of SJ components. Importantly, while membrane levels of the SJ core components Mega and ATPα were reduced within retromer clones (*Figure 2A–B*), levels of the SJ-associated TMP FasIII were not affected (*Figure 2—figure supplement 3*). These data mirror the phenotypes observed upon Shrub depletion (*Figure 1F–G* and *Figure 1—figure supplement 3*) and hint at a common pathway in apical delivery of a subset of SJ proteins.

Next, we quantified SJ density detected by TEM in wing discs depleted of Vps26 and found a reduction of electron-dense septa by about 60% (*Figure 2—figure supplement 4*). Consistent with compromised SJ integrity, a diffusible dye readily infiltrated Vps26-depleted tissue, indicating defective barrier function (*Figure 1—figure supplement 1*). We conclude that the retromer CSC regulates membrane levels of many SJ core components, thereby contributing to junction integrity and function.

In optical sections of *Vps35$^{MH20}$* clones, endogenously tagged Mega::YFP protein shows reduced overall levels throughout the mutant tissue (*Figure 2—figure supplement 3*). This suggests that misrouting of Mega into the degradative pathway may occur upon loss of *Vps35* function, as has been shown for many retromer cargos (*Franch-Marro et al., 2008*; *Harterink et al., 2011*; *Pocha et al., 2011*). Consistently, when retromer function and endo-lysosomal fusion are simultaneously impaired, Mega::YFP shows reduced SJ levels and accumulates in intracellular vesicles, hinting at a failure to be degraded (*Figure 2—figure supplement 1*). This result supports the idea that misrouting of Mega into the degradation pathway occurs when CSC function is compromised. We conclude that Mega behaves like a *bona fide* retromer cargo.

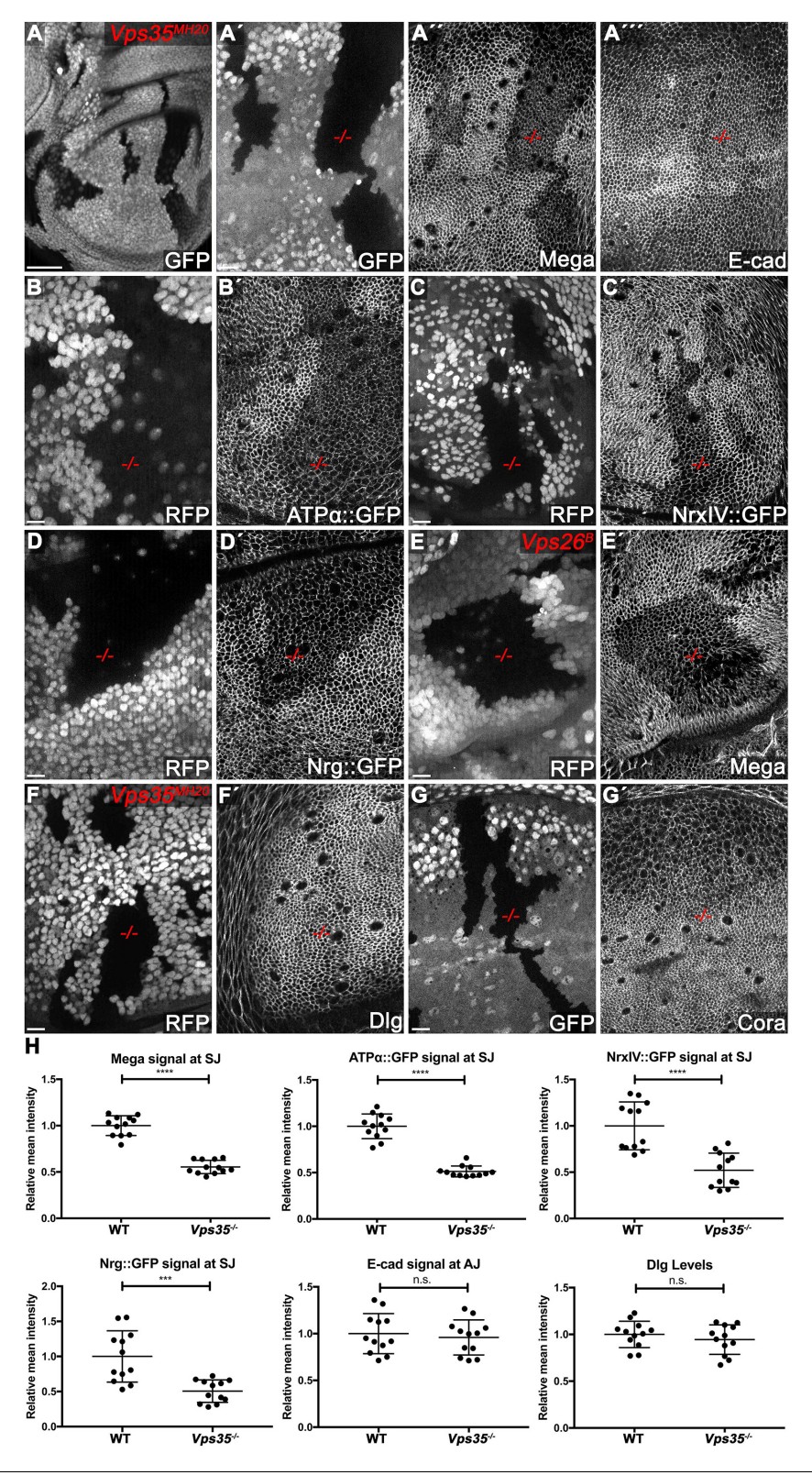

**Figure 2.** The retromer CSC regulates membrane levels of SJ core components. (**A**) Null mutant clones of the retromer component *Vps35* were generated in wing discs. Mutant tissue (-/-) is marked by absence of either GFP or RFP. Mega signal at the SJ is reduced in *Vps35^{MH20}* tissue (**A´**), while apical E-cad levels are unaffected (**A´´**). (**B–D**) Endogenously tagged SJ core components ATPα (**B**), NrxIV (**C**) and Nrg (**D**) show reduced membrane levels in *Vps35^{MH20}* tissue. (**E**) *Vps26* clones phenocopy *Vps35* clones with respect to junctional localization of Mega (compare **E´** with **A´**). (**F–G**)
*Figure 2 continued on next page*

*Figure 2 continued*

Junctional localizations of SJ-associated cytoplasmic proteins Dlg and Cora are unaffected in *Vps35^{MH20}* cells. (H) Quantification of junctional membrane levels of indicated proteins in wildtype and *Vps35^{MH20}* tissue. Each dot represents fluorescence intensity of a junctional region in either clonal (*Vps35^{MH20}*) or surrounding wildtype tissue. n = 4 discs for each component. A two-tailed t-test was used for statistical analysis with the significance levels **** representing a p-value of <0.0001, ***<0.001, and n.s. (not significant):≥0.05. Note that the transmembrane SJ components Mega, ATPα, NrxIV, and Nrg are similarly affected, showing reduced levels at the SJ within *Vps35^{MH20}* tissue by ~50%, while E-cad and Dlg apical levels do not significantly differ between wildtype and *Vps35^{MH20}* tissue. Scale bar in (A) 50 μm, in all other panels 10 μm.

The online version of this article includes the following source data and figure supplement(s) for figure 2:

**Source data 1.** Quantification of SJ protein apical levels in WT/Vps35 mutant clones.
**Figure supplement 1.** Mega is not target of lysosomal degradation under steady-state conditions.
**Figure supplement 2.** Retromer CSC-dependent transport of SJ components is conserved throughout tissues.
**Figure supplement 3.** Analysis of further SJ components in retromer CSC mutant clones.
**Figure supplement 4.** Retromer CSC function maintains SJ density.
**Figure supplement 4—source data 1.** Quantification of SJ density in WT and Vps26/retromer-depleted tissue.
**Figure supplement 5.** Retromer CSC-associated sorting nexins (SNX) are dispensable for SJ delivery of Mega.

To gain further insight into the CSC-dependent pathway required for SJ delivery of Mega, we investigated null mutant clones of known retromer-associated factors (SNX-BAR, Snx3, Snx27, and the WASH-complex component Fam21). Surprisingly, we failed to find any of these proteins to be required for the regulation of Mega membrane levels, raising questions on the mechanism governing CSC-dependent transport of Mega (*Figure 2—figure supplement 5*). Nevertheless, based on above findings, we propose a novel function of the retromer CSC in delivery of SJ core components to the junction, thereby contributing to SJ homeostasis in the proliferative wing disc epithelium.

## ESCRT regulates subcellular localization and mobility of the retromer CSC

The above results indicate that retromer is required for regulating membrane levels of Mega and other SJ components. Therefore, trapping of Mega in basally localized endosomal compartments upon *shrub*-RNAi expression could be a consequence of defective retromer-dependent endosomal export in this ESCRT deficient situation. We reasoned that loss of ESCRT function might alter organization and function of retromer-dependent carrier vesicles, thereby affecting cargo flux. To test this hypothesis, we analyzed the subcellular distribution of retromer components in Shrub-depleted cells.

We visualized the retromer CSC by using an endogenously tagged *Vps35* allele (Vps35::RFP, *Koles et al., 2016*). In anterior control compartments, Vps35::RFP was found in vesicular structures throughout the cell, but with significantly higher abundance in the apical cytoplasm (*Figure 3A–D*, arrows). Interestingly, this polarized apical localization is consistent with an apical transport hub at the junction level in wing disc cells that is characterized by enrichment of more than half of the *Drosophila* Rab GTPases (*Dunst et al., 2015*). In contrast to its concentration within the apical hub, little Vps35::RFP was detected in the basal cytoplasm of wildtype tissue (*Figure 3A′*). Upon *shrub*-RNAi expression however, Vps35::RFP strongly accumulated basally (*Figure 3A′*, arrowhead) while apical hub localization was almost completely abolished (*Figure 3A*). Therefore, loss of Shrub function appears to re-distribute Vps35::RFP positive vesicles from the apical hub into the basal cytoplasm where they accumulate. This apical to basal shift of retromer CSC positive vesicles is also evident in optical cross sections of wing discs expressing RNAi constructs for several ESCRT components (*Figure 3B–D*).

Importantly, RNAi directed against the ESCRT-I component TSG101 or Vps4, but not ESCRT-0, phenocopied *shrub*-RNAi expression with regard to Vps35::RFP subcellular localization (*Figure 3B–D*, *Figure 3—figure supplement 1*). This indicates that several ESCRT complexes are required for maintaining apical hub localization of the CSC in the wing disc epithelium. We found strong colocalization of Mega and Vps35::RFP on basal endosomal aggregates of *shrub*-RNAi expressing cells (*Figure 3—figure supplement 2*). This is in line with Mega being trapped in aberrant Rab7 and CSC positive endosomal compartments residing in the basal cytoplasm of ESCRT deficient cells. Our analysis of these basal aggregates revealed that besides containing Mega, they are enriched in ubiquitinated cargo and are coated with retromer components, Hrs and endosomal GTPases Rab5 and

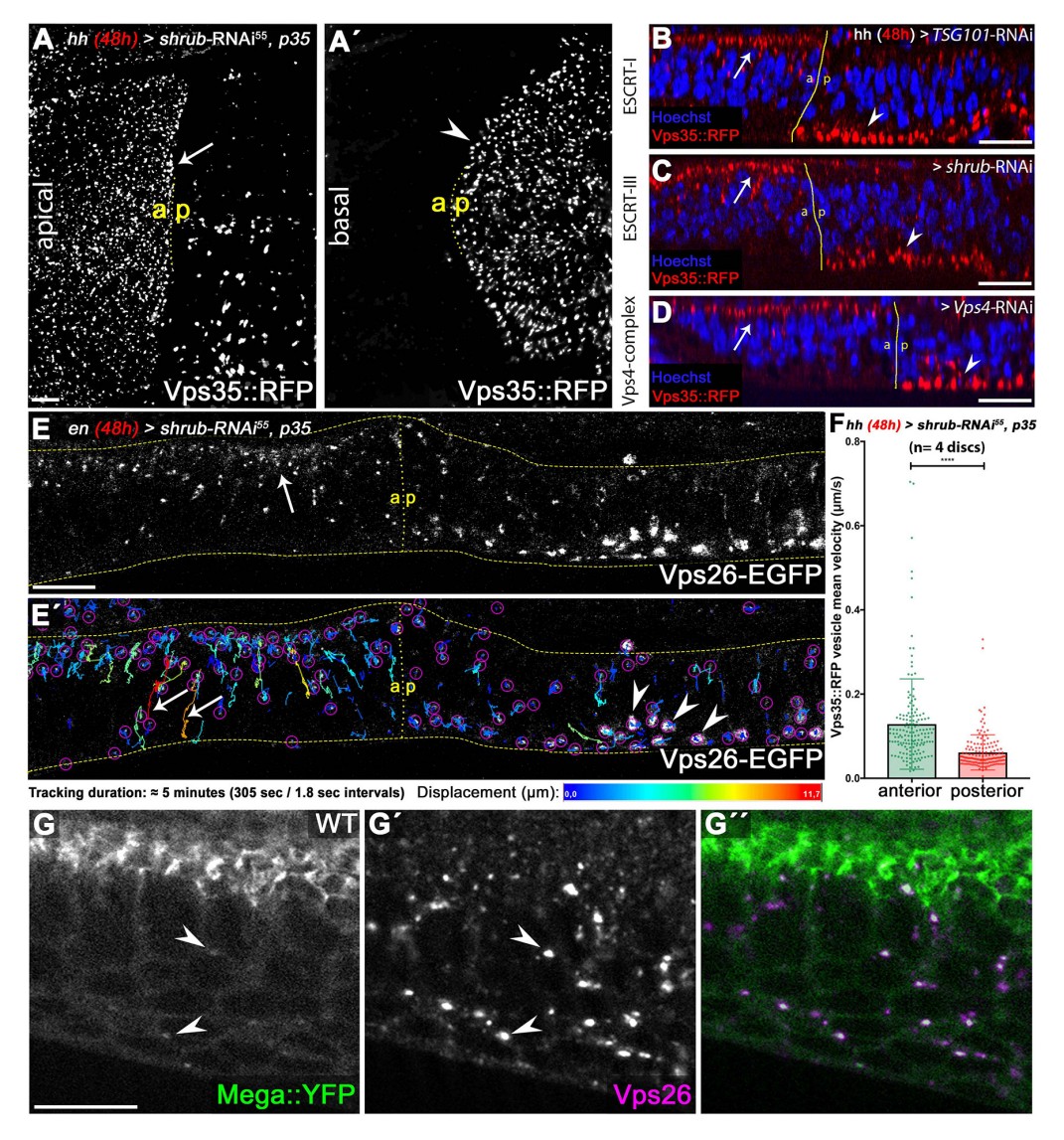

**Figure 3.** ESCRT function is required for CSC retromer apical hub localization and mobility. (**A**) Endogenously tagged Vps35::RFP localizes in a vesicular pattern within the apical hub in wildtype control compartments (arrow). After 48 hr of Shrub depletion, Vps35::RFP apical hub localization is largely abolished and basal accumulation is seen (arrowhead in **A'**). (**B–D**) The shift in subcellular localization of Vps35::RFP from the apical hub (wildtype, arrows) to basal aggregates (ESCRT-depleted compartment, arrowheads) is detected upon knockdown of ESCRT components TSG101, Shrub as well as Vps4. This indicates that regulation of retromer subcellular localization is a general ESCRT function in wing discs. (**E**) Live imaging and vesicle tracking of a genomic Vps26-EGFP construct at the anterior/posterior boundary of a wing disc expressing *shrub-RNAi* for 48 hr in the posterior compartment under control of *en*Gal4. Note the apical hub in the anterior control compartment and the large basally localized Vps26-EGFP aggregates in the posterior Shrub-depleted compartment (compare with Vps35::RFP in **C**). (**E'**) Individual vesicle trajectories visualize Vps26-EGFP movement within the tissue after a 5 min imaging acquisition. Note the high mobility of several vesicles along the apicobasal axis in the anterior control compartment (arrows). In contrast to long-distance shuttling of Vps26-EGFP in the control tissue, basal Vps26-EGFP aggregates (arrowheads) in the Shrub-depleted posterior compartment are largely static and overall vesicle dynamic is reduced. (**F**) Quantification of Vps35::RFP mobility in wildtype (anterior) and Shrub-depleted (posterior) tissue. Each dot represents mean velocity of a single tracked vesicle. Note the reduced mean velocity of Vps35::RFP and the high ratio of low mobility vesicles within Shrub-depleted compartments. A two-tailed t-test was used for statistical analysis with the significance level **** representing a p-value <0.0001. (**G**) Endogenously tagged Mega::YFP colocalizes with Vps26 in vesicular structures residing in the apical as well as basal cytoplasm (arrowheads). Scale bars in all panels represent 10 μm.

The online version of this article includes the following source data and figure supplement(s) for figure 3:

**Source data 1.** TrackMate raw data of Vps35::RFP vesicle tracking in WT and Shrub-RNAi +p35 tissue.

**Figure supplement 1.** ESCRT-0 is not required for SJ delivery of Mega.

**Figure supplement 2.** Characterization of basal 'class E-like' endosomal compartments induced by Shrub-RNAi.

*Figure 3 continued on next page*

*Figure 3 continued*

**Figure supplement 3.** TEM analysis of aberrant endosomal compartments induced by 48 hr of Shrub depletion.

Rab7 (*Figure 3—figure supplement 2*). We conclude that these aggregates likely represent *Drosophila* 'class E-like' compartments, which is supported by TEM analysis of basally localized endosomal structures within *shrub*-RNAi expressing wing disc tissue (*Figure 3—figure supplement 3*). While the regularly shaped and sized wildtype MVBs were absent in Shrub-depleted cells, we found a variety of enlarged, abnormal membranous compartments that are reminiscent of class E compartments in mammalian cells (*Doyotte et al., 2005*; *Stuffers et al., 2009*). We speculate that these irregular compartments aberrantly cluster endosomal machineries and cargos, thereby interfering with endosome function. Consequently, the SJ phenotype seen in *shrub*-RNAi tissue (*Figure 1*) could be explained by basal displacement of retromer-dependent carriers, which are required for targeting SJ components to the junctional region.

The apical bias of Vps35::RFP positive vesicles in wildtype cells suggests a polarized movement of CSC carriers along the apicobasal axis with relatively long dwell times at the apical hub. We reasoned that in wing discs, a majority of CSC-dependent cargo might be released preferentially at the apical pole of the cells. We turned to live imaging to study movement of CSC positive vesicles in wildtype and Shrub-depleted cells. We visualized CSC using Vps35::RFP or a transgene encompassing the genomic region of *Vps26* fused to a C-terminal EGFP-tag (*Wang et al., 2014*). In anterior control compartments, Vps26-EGFP localized in a vesicular pattern throughout the cells with increased abundance at the apical hub (*Figure 3E*, arrow). In the posterior Shrub-depleted cells however, Vps26-EGFP localization at the apical hub was reduced and basal accumulation was evident (*Figure 3E*). Therefore, live imaging of Vps26-EGFP recapitulates the apical to basal shift of Vps35::RFP in ESCRT-depleted fixed tissue (*Figure 3B–D*). We used Vps35::RFP/Vps26-EGFP to record time series of CSC positive carrier vesicles within wildtype and Shrub-depleted tissue. We applied the Fiji plugin TrackMate (*Tinevez et al., 2017*) to map individual trajectories of Vps35::RFP/Vps26-EGFP vesicles over a timeframe of 5 min (*Figure 3E'*). The results show that the CSC carriers in anterior control cells were highly mobile along the apicobasal axis with a subset of vesicles moving rapidly and relatively long distances between the apical and basal poles (*Figure 3E'*, arrows). This suggests that CSC carriers, although preferentially residing at the apical hub, are highly mobile and occasionally shuttle between the apical and basal poles of wing disc cells. In contrast, long-distance movement of Vps35::RFP/Vps26-EGFP vesicles was severely reduced in the posterior Shrub-depleted compartment (*Figure 3E' and F*). Importantly, the large basal CSC aggregates (class E-like compartments) were mostly immobile, showing no apparent movement along the apicobasal axis (*Figure 3E'*, arrowheads). These results indicate that ESCRT function is required for the mobility of CSC positive endosomes.

We hypothesize that certain retromer cargos might rely on CSC shuttling between the apical and basal poles for efficient transport. Mega::YFP vesicular structures overlapped extensively with Vps26 not only at the junctional level but also in vesicles with close proximity to the basal pole of the cells (arrowheads in *Figure 3G*). Colocalization analysis revealed that 71.9% of Mega::YFP vesicles along the apicobasal axis were positive for Vps26 (n = 3 discs/128 vesicles). This suggests that Mega and possibly other SJ components shuttle along the apicobasal axis in CSC positive carrier vesicles. We confirmed this by live imaging of Mega::YFP together with Vps35::RFP, which revealed extensive co-mobility of this retromer component with the vesicular fraction of Mega::YFP (*Animation 1*). Thus, Mega is moving along the apicobasal axis in CSC decorated vesicles.

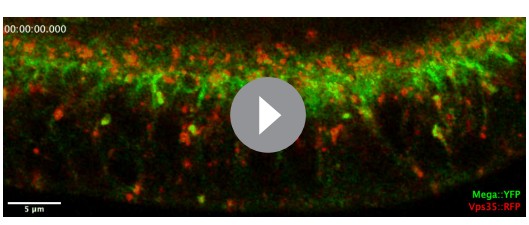

**Animation 1.** Live imaging time series of a wildtype wing imaginal disc expressing Mega::YFP and Vps35::RFP. Note the close association of Mega and Vps35 at dynamic vesicular structures along the apico-basal axis. Time stamp indicates passed time in minutes, starting from the initial frames of the acquisition.
https://elifesciences.org/articles/61866#video1

Together, the results reveal that ESCRT function regulates the mobility and apical hub localization of retromer CSC positive endosomes in wing imaginal discs. We conclude that by trapping CSC on aberrant endosomal compartments, loss of ESCRT function impairs retromer-dependent export of Mega from the endosome, consequently depleting its SJ pool in wing disc cells.

## Mega undergoes basal to apical transcytosis prior to reaching the SJ

Endosomal trafficking by retromer could occur via several different pathways. While certain retromer cargos are transported directly from endosomes to the plasma membrane, the classical retromer route involves cargo recycling in a detour via the Golgi. Besides these two endosomal recycling pathways, retromer has also been shown to regulate transcytosis from one membrane domain to another (*Vergés et al., 2004*). Prior to all retromer and ESCRT-dependent trafficking events, endocytosis of cargo is required for subsequent sorting within the endosomal system. We reasoned that blocking endocytosis of Mega might reveal the membrane domain from which it is internalized into the endosomal system and aid in understanding how the retromer pathway is involved in its trafficking.

We suppressed clathrin-mediated endocytosis by RNAi induced depletion of clathrin heavy chain (Chc) for 32 hr in the posterior compartment. While depletion of Chc did not have an apparent effect on junctional E-cad levels (*Figure 4A'*), Mega::YFP levels were strongly reduced at the SJ (*Figure 4A"*). Since membrane proteins accumulate at their site of endocytosis when the internalization process is inhibited, this result argues against Mega undergoing clathrin-dependent endocytosis at the apical membrane. Importantly, we found Mega::YFP accumulating at the most basal region of the lateral membrane (*Figure 4A"'*), suggesting that Mega is continuously removed from a basal membrane pool by endocytosis. These results also suggest that Mega, prior to accumulating at the SJ, undergoes clathrin-mediated endocytosis at the most basal section of the basolateral membrane (for the sake of simplicity, we will refer to this as the basodistal membrane).

Next, we interfered with dynamin-dependent endocytosis by expressing a dominant negative version of Shibire, the *Drosophila* dynamin homolog (UAS-*Shi$^{K44}$*, *Moline et al., 1999*) under the control of *hh*Gal4. An expression of Shi$^{K44}$ for 16 hr was sufficient to cause strong accumulation of Notch at the apical membrane, indicating that dynamin-dependent endocytosis of apically internalized membrane proteins is effectively impaired (*Figure 4B*). In line with diminished uptake of Notch into the endosomal system, we found reduced abundance of intracellular Notch vesicles in the posterior compartment (*Figure 4B*, arrowheads). In contrast to apically endocytosed Notch, Mega::YFP seemed to be reduced at the SJ level upon 16 hr of Shi$^{K44}$ expression (*Figure 4C*). In addition, it accumulated at the basal pole of the cells (*Figure 4C*, arrow). Reduced SJ levels accompanied by basodistal membrane accumulation of Mega::YFP were also found upon depleting the tissue of Rab5 by RNAi (*Figure 4D*). This common phenotype suggests that a critical step in SJ supply of Mega is its internalization from the basodistal membrane, hinting at a transcytosis-like mechanism with basal to apical direction.

We confirmed the close association of Mega with Chc in wing imaginal discs by detecting colocalization of Mega::YFP and Chc in vesicular structures in wildtype tissue (*Figure 4E*), which is consistent with Chc being part of the Mega proteome in *Drosophila* embryos (*Jaspers et al., 2012*). Importantly, Mega::YFP vesicles in proximity to the basodistal membrane were Chc positive, in line with Mega being endocytosed basally (*Figure 4E*, arrows). Colocalization analysis revealed that throughout the entire cell, 55.2% of Mega::YFP vesicles stained positive for Chc (n = 4 discs/87 vesicles). This Chc positivity rate was slightly increased to 63.01% in focal planes at the basal cell pole (n = 3 discs/73 vesicles), consistent with Mega::YFP entering the endosomal system by clathrin-dependent endocytosis at the basodistal membrane.

Together, these data support a transcytosis-like mechanism from the basodistal domain of the lateral membrane to the SJ, which is required to supply the junction with newly synthesized Mega. This process depends on the endocytic and early endosomal machinery as interference with dynamin, clathrin and Rab5 function leads to reduced SJ levels of Mega accompanied by its accumulation at the basodistal membrane. These results suggest that Mega needs to traverse through the endosomal system prior to reaching the SJ and argue against a conventional recycling function of retromer in this process. This notion is further supported by our comparative analysis of Mega trafficking to that of Crumbs (Crb), an established retromer recycling cargo in wing discs (*Pocha et al., 2011*). It

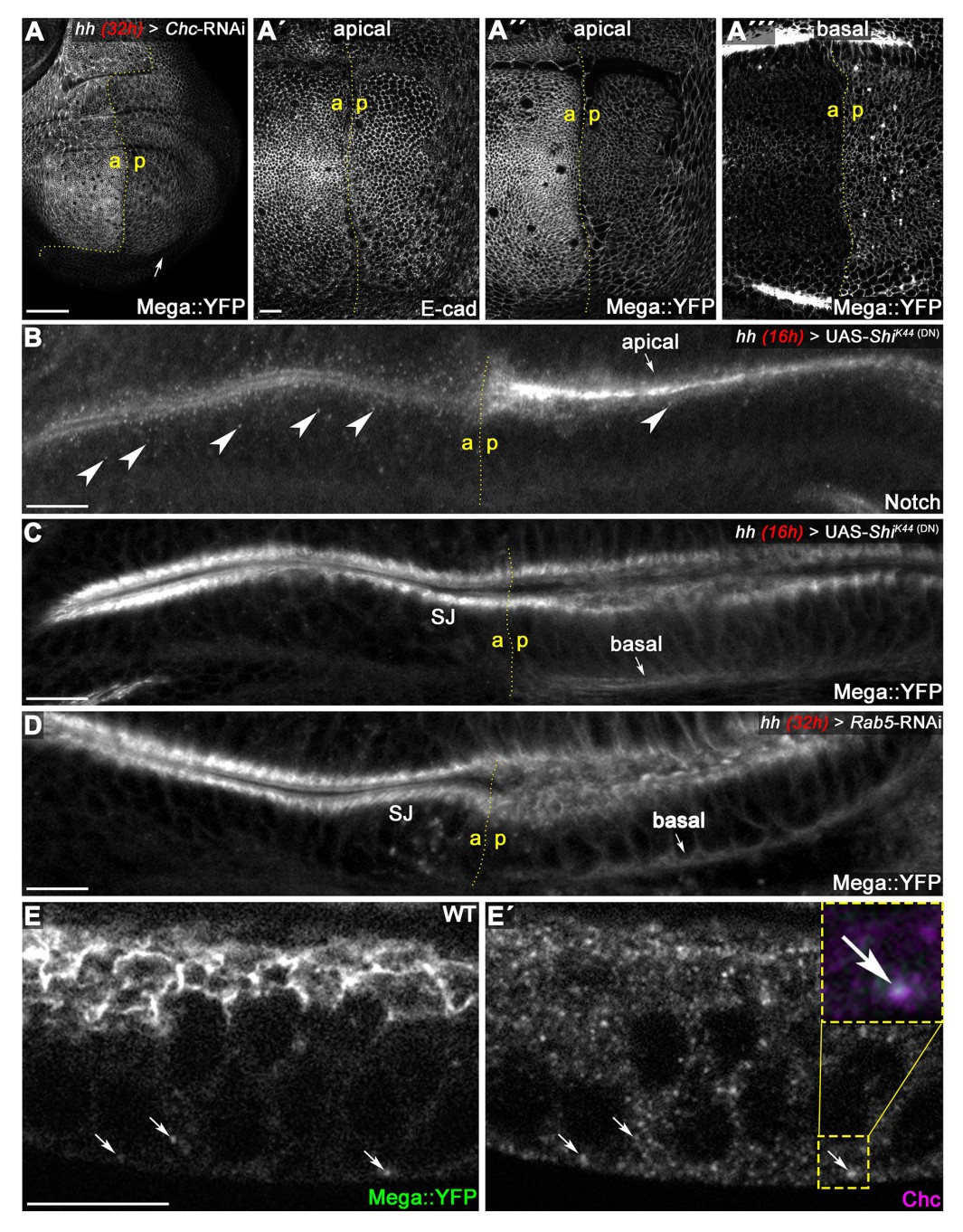

**Figure 4.** Mega is endocytosed at the basodistal membrane. (A) Depletion of clathrin via *Chc-RNAi* expression for 32 hr in the posterior compartment reduces apical membrane levels of Mega::YFP. While E-cad-based AJ appear unaffected following Chc depletion (A'), the junctional Mega::YFP pool is reduced (A''). In a basal focal plane, Mega::YFP is seen accumulating in the basodistal part of the lateral membrane in the *Chc-RNAi* expressing posterior compartment. (B–C) Expression of a dominant negative Dynamin construct (Shi$^{K44}$) leads to apical accumulation of Notch within an epithelial fold of the hinge region (arrow in B) and reduces the amount of Notch positive vesicles (arrowheads). In contrast, posterior Mega::YFP signal at the SJ is reduced and a basal pool is visible (arrow in C). (D) Depleting Rab5 in the posterior compartment yields a similar Mega::YFP phenotype with respect to junctional levels and basal accumulation (arrow in D). (E) In line with internalization from the basodistal membrane, Mega::YFP colocalizes with clathrin at vesicular structures near the basal cell pole and along the apicobasal axis (arrows in E and E'). Scale bars in all panels represent 10 µm except for (A), in which it represents 50 µm.

The online version of this article includes the following figure supplement(s) for figure 4:

**Figure supplement 1.** Comparison of Mega phenotypes with the retromer CSC cargo Crumbs.

revealed distinct phenotypes upon Vps35 loss or inhibition of endocytosis, suggesting that these proteins do not traverse a common retromer-dependent pathway (*Figure 4—figure supplement 1*).

We conclude that Mega is not undergoing apical recycling but rather requires basal to apical transcytosis for junctional delivery. Thus, our data suggest that transcytosis of SJ components is not limited to initial SJ formation in the embryo (*Tiklová et al., 2010*), but is also required in a proliferative epithelium to maintain the junctional pool of the SJ component Mega.

## Basal to apical transcytosis of Mega depends on ESCRT and retromer function

The complex anterograde trafficking of Mega to the SJ prompted us to investigate it in more detail. We generated an HA-tagged UAS-Mega construct, allowing us to analyze its delivery to the SJ upon overexpression. Continuous *hh*Gal4-driven expression of UAS-HA-Mega led to its integration into SJ, which we confirmed by colocalization with Lac::GFP (*Figure 5A*"). Interestingly, in contrast to endogenous Mega, we detected a fraction of HA-Mega at the most basal part of the epithelium (*Figure 5A*). This suggests that the basodistal membrane pool of Mega that we observed upon endocytosis block is also detectable upon Mega overexpression. We also found increased abundance of intracellular vesicles, consistent with elevated trafficking of Mega within the cells (*Figure 5B*). HA-Mega colocalized with Vps35::RFP on vesicles (*Figure 5B*, arrows), reminiscent of endogenous Mega::YFP shuttling in Vps35::RFP positive endosomal carriers (*Animation 1*). Hence, we conclude that over-expression of UAS-HA-Mega recapitulates the hallmarks of Mega intracellular transport.

While the majority of HA-Mega signal upon continuous overexpression was detected at the SJ level (*Figure 5C*, arrowhead), HA-Mega was also found in lateral and basal membrane regions (*Figure 5C*, arrow). Strikingly, the basodistal pool of HA-Mega was characterized by an intense membrane localization pattern resembling the staining of the junctional region (*Figure 5C'*).

The previous experiments suggested that Mega is continuously endocytosed from the basodistal membrane to supply the SJ pool. We therefore reasoned that basodistal membrane accumulation might be a short-lived intermediate step in Mega transport toward the SJ. To proof this assumption, we conducted a pulse-chase experiment using the Gal4/Galt80ts system and found that HA-Mega localization was much more confined to the SJ (similar to endogenous Mega localization) when its expression was halted for 14 hr after continuous expression (*Figure 5D*). After this chase, the basodistal membrane pool was almost completely diminished (*Figure 5D'*). This is consistent with a transient localization of Mega at the basodistal membrane prior to endocytosis and subsequent targeting to the SJ.

Next, we investigated the roles of ESCRT and retromer in HA-Mega trafficking. Strikingly, when we expressed HA-Mega in *shrub*-RNAi tissue, it failed to reach the SJ altogether and instead was trapped within aberrant endosomal compartments in the basal cytoplasm (*Figure 5E*, arrow). Colocalization with Vps35::RFP confirmed that HA-Mega accumulated in basal retromer CSC positive compartments (*Figure 5F*, arrows), consistent with the results obtained from Mega staining in *shrub*-RNAi tissue (*Figure 3—figure supplement 2*). When we expressed HA-Mega in CSC-depleted tissue (*Vps35*-RNAi), a similar, albeit weaker trapping of HA-Mega in basally localized vesicular compartments was observed (*Figure 5G*, arrows). In contrast to exclusive localization of HA-Mega in basal aggregates upon *shrub*-RNAi expression (*Figure 5F*, arrows), we also detected a pool of HA-Mega at the SJ level (*Figure 5G*). However, imaging the SJ plane revealed that apical HA-Mega was mostly vesicular with no apparent membrane staining (*Figure 5I*). This is in strong contrast to HA-Mega localization in wildtype tissue, characterized by a distinct junctional honeycomb pattern in both the disc proper and the overlaying peripodial disc cells (*Figure 5J*). These results suggest that HA-Mega fails to integrate into the SJ in retromer or ESCRT-depleted wing disc cells.

Recognition of proteins by the ESCRT machinery requires cargo ubiquitination, serving as a sorting signal for the ILV pathway (*Bilodeau et al., 2002*; *Katzmann et al., 2001*; *Shih et al., 2002*). To test whether HA-Mega requires ubiquitination for its integration into SJ, we devised a UAS-HA-Mega construct in which all intracellular lysines (K) were changed to arginines (R) (UAS HA-Mega$^{K2R}$). This construct should be devoid of any potential ubiquitination, thereby preventing a possible recognition by ESCRT and sorting into the ILV pathway. However, upon expression in wing disc cells, HA-Mega$^{K2R}$ targeting to the junctional membrane was not impaired and the subcellular localization of the construct was indistinguishable from wildtype HA-Mega (*Figure 5—figure supplement 1*). This

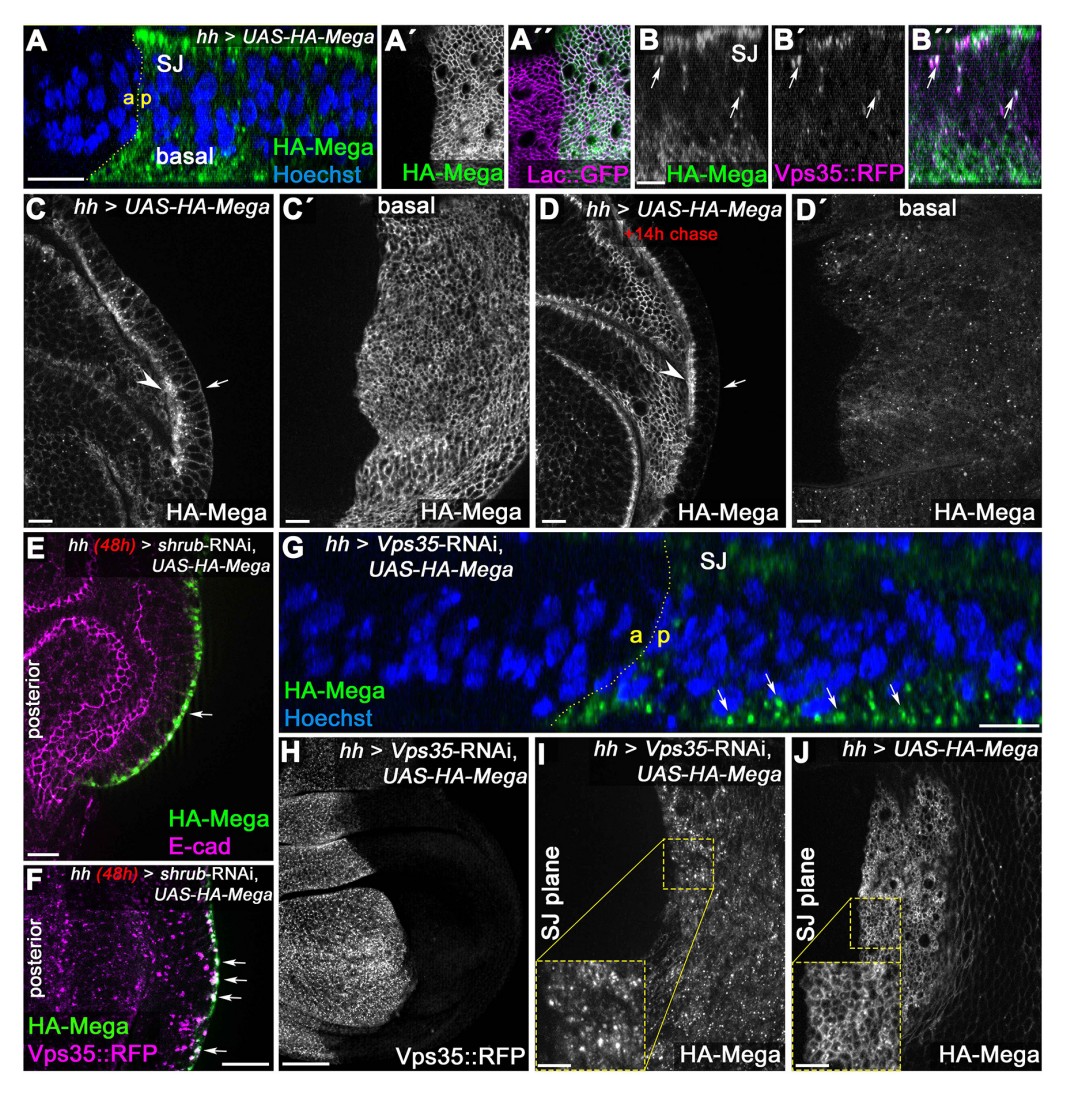

**Figure 5.** Basal to apical transcytosis of Mega requires ESCRT and retromer function. (**A**) UAS-driven overexpression of HA-tagged Mega reveals a basal pool opposing the apical SJ fraction. Similarly to endogenously tagged Mega::YFP (see *Figure 3G*), HA-Mega colocalizes with the CSC on vesicles along the apicobasal axis (arrows in **B**). (**C**) Continuous overexpression of HA-Mega yields a strong SJ signal (arrowhead) as well as lateral and basal (arrow) staining of HA-Mega. (**C′**) In a basal focal plane, the strong accumulation of HA-Mega in a junction like fashion at the basodistal membrane is seen. (**D**) Following a 14 hr chase at 18°C to halt further expression, HA-Mega signal is almost exclusively confined to the SJ (arrowhead) with very little basal staining (arrow). (**D′**) The basodistal membrane pool of HA-Mega (compare with C′) vanished after 14 hr chase, in line with a transient localization of HA-Mega at this membrane domain. (**E**) In Shrub-depleted tissue, overexpressed HA-Mega does not reach the SJ and is found exclusively in large basal aggregates (arrow). (**F**) These basal aggregates are positive for Vps35::RFP (arrows). (**G**) Similarly, in Vps35-depleted tissue, apical SJ signal of HA-Mega is faint and vesicular accumulation in the basal cytoplasm is seen (arrows). (**H**) Knockdown efficiency of the *Vps35*-RNAi. (**I**) No distinct membrane staining of HA-Mega is visible in the SJ plane of Vps35-depleted tissue, in contrast to HA-Mega junctional localization in wildtype tissue (**J**). Scale bar in (**B**) represents 5 μm, in (**G**): 50 μm and in all other panels: 10 μm.

The online version of this article includes the following figure supplement(s) for figure 5:

**Figure supplement 1.** Mega transport toward the SJ is independent of intracellular lysines.

result suggests that Mega transcytosis and SJ integration does not require ubiquitination and that direct interaction with ESCRT components and sorting into ILVs is dispensable during anterograde transport of Mega. These data are compatible with a model of indirect regulation of Mega trafficking by ESCRT that is based on the requirement for ESCRT function in maintaining apical hub localization and mobility of the retromer CSC.

Based on above findings, we conclude that a functional endosomal system is critically required for Mega transcytosis from the basodistal to the apical membrane, which is an essential prerequisite for its delivery to the SJ. Importantly, our data indicate a crucial role for ESCRT and retromer in a pathway that regulates anterograde trafficking of an apically localized membrane protein.

## Mega translation and subsequent exit from the Golgi occurs in the basal cytoplasm

The previous experiments suggested that Mega, despite residing in apical SJs, requires crucial trafficking steps at the basodistal plasma membrane prior to its integration into the junction. We therefore wondered whether locally restricted synthesis of Mega in the basal part of the cell might account for the transient pool at the basodistal plasma membrane observed upon endocytosis block or Mega overexpression.

We used RNA fluorescence in situ hybridization to reveal the subcellular localization of Mega transcripts. To verify RNA probe specificity, we expressed UAS-HA-Mega using *hh*Gal4, which should strongly increase transcript abundance and consequently fluorescence signal intensity in the posterior compartment. Consistently, we found intense Mega RNA staining in the posterior compartment, confirming the specificity of our probe (*Figure 6A*). We also detected some apical fluorescence signal that is notably visible in the epithelial fold of the hinge region (*Figure 6A,B*, red asterisk). Very likely, this fraction represents unspecific signal from fluorophores accumulating in the extracellular space between the peripodial membrane and the disc proper, since the signal intensity did not differ between the wildtype anterior and the HA-Mega overexpressing posterior compartment (*Figure 6B*, red asterisk). In contrast, the specific intracellular signal representing Mega transcript was markedly increased in the posterior compartment (*Figure 6B*). Strikingly, endogenous, as well as overexpressed Mega RNA, was almost exclusively detected in dotted structures residing in the basal cytoplasm (*Figure 6B*, arrows and arrowheads, respectively).

The basal subcellular localization of Mega RNA is in strong contrast to that of the protein, which almost exclusively localizes in SJs at the apical pole of the cells (*Figure 6C*). Thus, the discrepancy in RNA and protein localization may explain the necessity for a transcytosis route from the basodistal membrane to the SJ. We reasoned that Mega mRNA is translated basally, which may result in its initial targeting to the basodistal membrane following passage through the Golgi. Consistent with this idea, we found robust colocalization of HA-Mega with the Golgi marker Golgin84 (*Riedel et al., 2016*) at the basal pole of the cells (*Figure 6D*, arrows). In contrast, little colocalization was found in the apical region in proximity to the SJ. This is surprising, since the majority of Golgi stacks appeared to reside apically (*Figure 6D'*). To further study the early biosynthetic trafficking of Mega, we induced a short *hh*Gal4-driven expression of UAS-HA-Mega for 2 hr. Only single cells within the posterior compartment showed detectable expression of HA-Mega (*Figure 6E*, arrows). Strikingly, we occasionally detected cells containing exclusively basal vesicular HA-signal that overlapped with Golgin84 (*Figure 6F*, arrows). Other cells showed extensive spreading of HA-Mega along the apico-basal axis and colocalization with Lac::GFP at the apical membrane, indicating integration into the SJ (*Figure 6G–I*, arrow). These data are consistent with the first appearance of HA-Mega post biosynthesis in basal Golgi stacks, followed by transport to the basodistal membrane and subsequent transcytosis toward the SJ.

Together, the results suggest that basal subcellular localization of Mega mRNA supports its local translation and subsequent secretion of Mega protein from basal Golgi stacks. Thus, the transient basodistal membrane pool of Mega is likely a consequence of this locally restricted secretion and represents newly synthesized protein that has not yet entered the transcytosis pathway required for anterograde delivery to the SJ.

## Discussion

Here, we describe a novel role of the endosomal ESCRT and retromer machineries in regulating SJ maintenance in a proliferating epithelium, the wing disc of *Drosophila*. By studying the intracellular trafficking route of the SJ component Mega, we reveal a transcytosis-like mechanism in a basal to apical direction that delivers Mega to the junctional region and requires ESCRT and retromer functions.

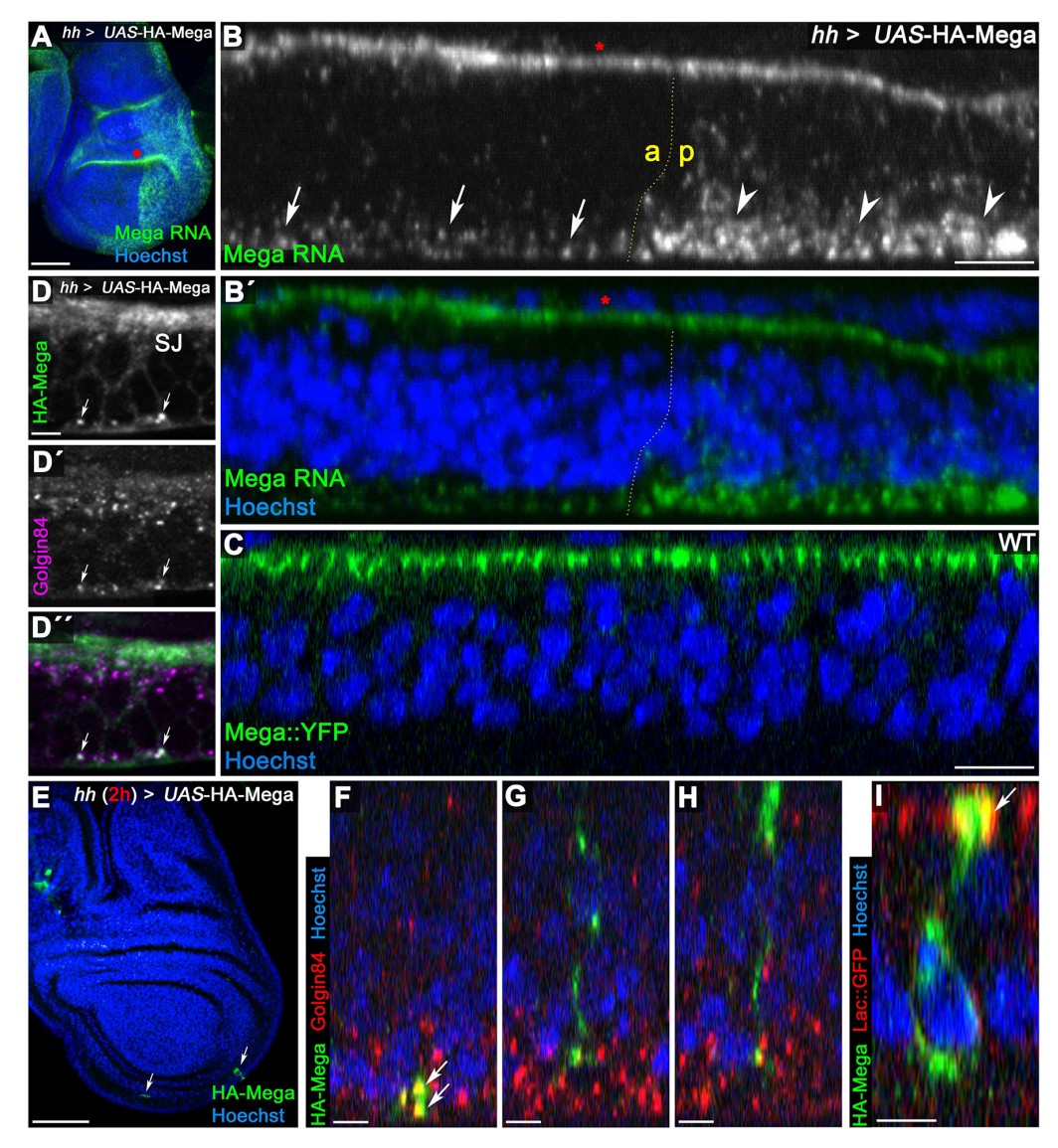

**Figure 6.** Mega is secreted from Golgi stacks within the basal cytoplasm. (A) Fluorescence in situ hybridization (FISH) signal from a Mega RNA probe after posterior overexpression of UAS-HA-Mega. Note the strong posterior signal, indicating probe specificity. Unspecific signal between peripodial membrane and disc proper is visible in the epithelial fold of the hinge region (red asterisks in **A** and **B**). (B) Mega RNA is detected primarily in the basal cytoplasm of both the wildtype and the UAS-HA-Mega overexpressing posterior compartment (arrows and arrowheads, respectively). Note the higher abundance of fluorescence signal in the posterior compartment with a subcellular localization comparable to endogenous Mega RNA within the anterior tissue. (C) In stark contrast to basally localized mRNA, Mega protein (Mega::YFP) localizes within the SJ at the apical pole of the cells. (D) Overexpressed Mega colocalizes with the Golgi resident protein Golgin84 in vesicular structures within the basal cytoplasm (arrows), consistent with basal secretion of Mega. Note the high abundance of Golgi stacks in proximity to the SJ but lack of overlap with HA-Mega in this apical region. (E–I) A short expression of HA-Mega for 2 hr yields discs that contain single cells with detectable HA-Mega expression (arrows). (F) Some cells stain for HA-Mega exclusively in basal, punctate spots which overlap with Golgin84 (arrows). This is in line with initial appearance of HA-Mega at the basal cell pole during secretion from local Golgi stacks. (G–H) Intracellular spreading of HA-Mega along the apicobasal axis toward the SJ. (I) HA-Mega extensively colocalizes with Lac::GFP at the apical pole, indicating efficient SJ targeting (arrow). Scale bars in (A, B) represent 50 µm, in (B), (C): 10 µm and in all other panels: 5 µm.

While transcytosis of SJ components has been shown to occur during the initial establishment of the SJ in the embryo (*Tiklová et al., 2010*), we here reveal that this mechanism is also continuously required during maintenance of the SJs in a rapidly proliferating epithelium. Our data reveal that the retromer CSC functions downstream of ESCRT to export Mega from the endosome. We propose a

novel physiological role for the retromer CSC in regulating membrane levels of several SJ core components. While our data suggest that retromer fails to export Mega from aberrant endosomes induced by ESCRT depletion, the exact mechanism behind this remains to be determined.

## Requirement for ESCRT function in CSC-dependent endosomal retrieval of Mega

Our data reveal a critical requirement for ESCRT in a transport pathway that depends on retromer-mediated transcytosis to deliver newly synthesized Mega to its apical destination. Defects in endosomal retrieval upon ESCRT inactivation have been previously described in other systems, such as yeast or mammalian cells and, thus, appear to represent a common feature of the pleiotropic ESCRT deficient phenotype. In yeast, the endosome-to-Golgi retrieval of the sorting receptor Vps10p and its cargo carboxypeptidase Y (CPY) depends on retromer function (*Seaman et al., 1997*; *Seaman et al., 1998*). ESCRT mutant strains accumulate CPY in class E compartments from which retrieval to the Golgi is blocked (*Babst et al., 1997*; *Piper et al., 1995*; *Raymond et al., 1992*). Similarly, the mammalian retromer cargo mannose 6-phosphate receptor (M6PR) also failed to recycle from endosomes to the Golgi in HeLa cells depleted of TSG101/ESCRT-I function (*Doyotte et al., 2005*). In this study, the authors suggested that generation of class E compartments occurs at the expense of endosomal tubules (*Doyotte et al., 2005*). Consistently, the retromer-associated tubulation factor SNX1 and its yeast homolog Vps5p were found on the rims of mammalian and yeast class E compartments, respectively (*Doyotte et al., 2005*; *Seaman et al., 1998*). Together with our finding of CSC accumulation on *Drosophila* class E-like compartments (*Figure 3*, *Figure 3—figure supplement 2*), this suggests that ESCRT deficient endosomes remain coated with retromer components but fail to export specific cargo.

While we cannot rule out the possibility of ESCRT components directly cooperating with retromer to form recycling tubules (note that SNX-BAR, Snx3, and Snx27 are not required for Mega transport; see *Figure 2—figure supplement 5*), we favor an indirect mechanism linking ESCRT and retromer in this transport pathway. Our analysis of the aberrant endosomal compartments induced upon Shrub depletion revealed that they are enriched in endosomal organizers such as Rab5 and Rab7, which could potentially interfere with retromer-dependent export when their activity at the limiting membrane is unrestrained (*Figure 3—figure supplement 2*). While the role of Rab7 in endosomal recruitment of the CSC is well established, the necessity for Rab7 GDP/GTP cycling during retromer-dependent carrier generation is still under debate (*Jia et al., 2016*; *Jimenez-Orgaz et al., 2018*; *Seaman et al., 2009*). Rab7 and its GTPase-activating protein (GAP) Tbc1d5 are interaction partners of the CSC and can modulate its capability to retrieve endosomal cargo (*Jia et al., 2016*; *Seaman et al., 2009*). For example, interfering with Rab7-GTP hydrolysis by Tbc1d5 depletion yielded defects in retromer-dependent transport in HeLa cells (*Jia et al., 2016*). Strikingly, under these conditions, retromer cargo was trapped in CSC-coated endosomes, paralleling our observation of Mega subcellular localization upon ESCRT depletion (*Jia et al., 2016*). Similarly, by exposing the interplay between the CSC component Vps29, Tbc1d5, and Rab7 in adult *Drosophila* brains, the authors of a recent study reported the capability of endosomal Rab7 to interfere with retromer CSC function in vivo (*Ye et al., 2020*).

While the exact mechanism rendering retromer dysfunctional at *Drosophila* class E compartments remains to be determined, our data support the mounting pool of evidence that ESCRT is required for multiple endosomal retrieval pathways. It is therefore likely that aspects of the pleiotropic ESCRT phenotype in metazoans stem from defective export of proteins from the endosomal system. For example, in *Drosophila*, leaky SJ could support ESCRT-mediated neoplastic transformation by permitting diffusion of signaling molecules within the imaginal disc tissue.

## Transcytosis as a means to provide biosynthetic delivery of SJ components

We here found that biosynthetic delivery of Mega depends on a transcytosis-like mechanism from the basodistal to the apical plasma membrane. This long-distance transport required sequential action of endocytic (clathrin, dynamin, Rab5) and endosomal (ESCRT, retromer CSC) machineries. Importantly, the finding that overexpressed HA-Mega is unable to reach the SJ in absence of retromer and ESCRT function (*Figure 5*) is in agreement with biosynthetic delivery of Mega relying on

endosomal function. Therefore, although we cannot exclude the possibility that Mega transiently passes the Golgi after endocytosis at the basodistal membrane, we favor the unconventional transcytosis model. Strikingly, while retromer-dependent endosomal recycling has been extensively documented, only one mammalian cell culture study implicated retromer in transcytosis from one membrane domain to another (*Vergés et al., 2004*). Thus, SJ delivery of Mega in imaginal discs represents a novel physiological role of retromer to study this process in vivo. The finding that CSC-mediated anterograde transport of Mega is independent of retromer-associated sorting nexins (*Figure 2—figure supplement 5*) indicates that this transcytosis pathway is distinct from many established CSC-dependent routes and suggests that it may require unknown cofactors (or does not require endosomal tubulation).

Our analysis of *Vps35* clones in pupal wings or leg imaginal discs revealed that in these tissues, clones completely devoid of the SJ core component NrxIV occur frequently (*Figure 2—figure supplement 2*). Similarly, *shrub* mutant clones in the pupal notum were entirely lacking junctional ATPα (Roland Le Borgne, personal communication, July 2020). This is in contrast to surface levels of SJ components in *Vps35* mutant wing discs, which were consistently reduced by about 50% (*Figure 2*). This provokes the hypothesis that a parallel endosomal export pathway for SJ components may exist in wing discs that could partially compensate for loss of retromer. However, we think this is unlikely since overexpressed HA-Mega fails to reach the SJ not only upon Shrub but also upon Vps26 depletion (*Figure 5E–J*). One has to keep in mind that this experiment specifically monitors delivery of newly synthesized HA-Mega while the *Vps35* clonal analysis assesses the impact of retromer loss of function on pre-existing SJ. Thus, in a clonal situation, a 'thinning out' of junctions is expected with consecutive rounds of cell division, which could explain incomplete phenotypic expressivity in wing disc *Vps35* clones. Nevertheless, it remains to be determined why in leg discs and pupal wings, SJ appear to be more sensitive toward CSC loss (*Figure 2—figure supplement 2*). During metamorphosis, wing imaginal disc cells undergo drastic morphogenetic changes to form the pupal wing epithelium, a process known to require AJ remodeling (*Classen et al., 2005*). It is therefore possible that analogously to AJ, SJ may also be actively remodeled during pupal wing formation. This could explain the strong requirement for retromer function in maintaining SJ integrity in a tissue undergoing morphogenetic changes.

Paralleling our findings described here, a previous study showed that embryonic SJ formation depends on endocytosis and subsequent redistribution of junction components from the lateral membrane to the SJ (*Tiklová et al., 2010*). Thus, transcytosis of SJ components is a mechanism likely required for both initial SJ formation as well as maintenance of SJ integrity in non-embryonic tissues in *Drosophila*. While we did not assess the roles of ESCRT and retromer CSC in initial SJ formation, it is conceivable that they are already required for transporting SJ components during embryogenesis. Consistently, *shrub* mutant embryos display a defective epithelial barrier function, suggesting they fail to form functional SJ (*Dong et al., 2014*).

The reason for SJ maintenance to rely on such an elaborate trafficking of its components remains to be determined. It has been suggested that SJ components form stable complexes prior to integration into the junction. Potentially, essential post-translational modifications of certain SJ components required for complex formation may occur exclusively at the basodistal membrane or during the passage through the endosomal system. If transient localization of SJ components at the basodistal membrane is a prerequisite for efficient SJ core complex formation, depositing transcripts for structural components such as Mega in the basal cytoplasm would shorten the route individual SJ components need to pass prior to SJ complex formation at the basodistal membrane domain. Alternatively, the finding that Mega mRNA predominantly localizes in the basal cytoplasm (*Figure 6*) provides the foundation for another hypothesis: It is widely accepted that apical and basolateral cargos undergo motif-based sorting leading to secretion toward the respective membrane domains (*Stoops and Caplan, 2014*). Basal subcellular localization of Mega transcripts could potentially reflect an apical/basolateral sorting divergence at the mRNA level. Accordingly, basal translation and exocytosis of Mega (induced by a putative basal/basolateral sorting signal) may lead to its targeting toward the basodistal membrane, despite the fact that the SJ resides apically in wing disc cells. Thus, transcytosis may serve as an adaptation for redistribution of cargos to their destined membrane domain when they are initially secreted to a different one due to early sorting signals. Since the columnar wing imaginal disc cells have a very elongated shape and possess Golgi stacks all along the apicobasal axis, it is conceivable that certain Golgi stacks residing at the apical and basal

poles are specialized in secreting apical and basolateral cargos, respectively. Although this is highly hypothetical, systematic analysis of transcript localization for apical (e.g. E-cad) and basolateral (e.g. SJ components) cargos could reveal a potential spatial separation of distinct secretory routes already at the mRNA level.

In wing imaginal discs, a similarly complex transcytosis route (albeit with an apical to basal direction) has been described for the signaling molecule Wingless, which is translated apically, transiently presented at the apical membrane and finally transcytosed toward the basal membrane where it is secreted (*Yamazaki et al., 2016*). Thus, distinct transcytosis pathways in the wing disc epithelium provide a mechanism for targeting certain proteins to their site of action, specifically when the protein is translated far away from its terminal destination.

## A novel retromer function in transport of SJ proteins

In this study, we unravel a novel physiological retromer function in regulating surface levels of a claudin and other structural SJ components (e.g. Nrg, ATPα, Lac, NrxIV, Cont) in several *Drosophila* tissues (*Figure 2* and *Figure 2—figure supplement 2*). Presently, we do not know how the SJ components are selected for this retromer-dependent pathway, and whether it requires physical interaction with CSC components. Since SJ proteins may traverse the endosomal system in complex, the vast number of different components brings about a plethora of possible interaction sites. Importantly, a mass-spectrometry-based study of the Mega interactome did not detect any retromer CSC components or associated factors but confidently found SJ core components as well as clathrin (*Jaspers et al., 2012*). While the interaction mode of CSC and SJ proteins remains to be determined, our data reflect the assumption that several SJ core components represent novel putative retromer cargos in *Drosophila*.

Strikingly, among the proteins affected by retromer loss of function, many possess mammalian homologs (e.g. NrxIV/CNTNAP2, ATPα/ATP1A1, Nrg/NRCAM). This suggests they could represent a novel set of conserved retromer cargos. Indeed, several lines of evidence suggest that ESCRT/retromer-mediated transport of SJ components may be evolutionary conserved from *Drosophila* to mammals. Depletion of ESCRT-I component TSG101 in mammalian epithelial cells led to a reduction of trans-epithelial resistance (TER), indicating defects in TJ-mediated barrier function (*Dukes et al., 2011*). Additionally, Claudin-1, an essential TJ component, continuously underwent endocytosis and recycling back to the plasma membrane in several mammalian cell lines in a process requiring ESCRT function (*Dukes et al., 2011*).

Importantly, the mechanism behind reduced recycling and entrapment of Claudin-1 in ubiquitin-positive aberrant endosomes upon interference with ESCRT function remained elusive (*Dukes et al., 2011*). Thus, it is unknown how the export of Claudin-1 from the endosomal system in mammalian cells is achieved (*Dukes et al., 2011*). By revealing an ESCRT-dependent function of the retromer CSC in claudin endosomal export in *Drosophila*, our data may provide an explanation for a possibly conserved trafficking pathway of claudins. In support of this, Claudin-1 and Claudin-4 membrane levels were significantly reduced in a mass spectrometry-based surface proteome study of Vps35-depleted human cells (*Steinberg et al., 2013*). Furthermore, the TJ protein Zonula occludens-2 (ZO-2) was strongly enriched in a Vps26 interactome, suggesting that a presumptive retromer function in TJ maintenance in mammalian cells may not be limited to claudins, similar to our findings in *Drosophila* presented here (*Steinberg et al., 2013*). It remains to be determined whether a physiological role of retromer in mammalian TJ maintenance occurs also in vivo. An increasing amount of tools, such as conditional *Vps35* knockout mice (*de Groot et al., 2013*), will enable analysis of this putatively conserved retromer function in mammalian systems and reveal any possible implications in development and/or disease.

## Materials and methods

### Key resources table

| Reagent type (species) or resource | Designation | Source or reference | Identifiers | Additional information |
| --- | --- | --- | --- | --- |

*Continued on next page*

*Continued*

| Reagent type (species) or resource | Designation | Source or reference | Identifiers | Additional information |
|---|---|---|---|---|
| Genetic reagent (*D. melanogaster*) | UAS-HA-Mega (II.) and (III.) | This paper | | N-terminally HA-tagged WT Mega for Gal4/UAS-driven expression |
| Genetic reagent (*D. melanogaster*) | tubP-HA-Mega (II.) | This paper | | N-terminally HA-tagged WT Mega expressed under the control of a tubulin promotor |
| Genetic reagent (*D. melanogaster*) | UAS-HA-Mega$^{K2R}$ | This paper | | N-terminally HA-tagged Mega (intracellular lysines exchanged with arginines) for Gal4/UAS-driven expression |
| Genetic reagent (*D. melanogaster*) | UAS-Rab5-RNAi | Vienna *Drosophila* Resource Center | V34096 | |
| Genetic reagent (*D. melanogaster*) | UAS-Vps35-RNAi | Bloomington *Drosophila* Stock Center | BL22180 | |
| Genetic reagent (*D. melanogaster*) | UAS-Chc-RNAi | Vienna *Drosophila* Resource Center | V103383 | |
| Genetic reagent (*D. melanogaster*) | UAS-Vps4-RNAi | Vienna *Drosophila* Resource Center | V101722 | |
| Genetic reagent (*D. melanogaster*) | UAS-TSG101-RNAi | Bloomington *Drosophila* Stock Center | BL35710 | |
| Genetic reagent (*D. melanogaster*) | UAS-shrub-RNAi | *Sweeney et al., 2006* | | |
| Genetic reagent (*D. melanogaster*) | UAS-Vps26-RNAi | Bloomington *Drosophila* Stock Center | BL38937 | |
| Genetic reagent (*D. melanogaster*) | UAS-Vps39-RNAi | Bloomington *Drosophila* Stock Center | BL42605 | |
| Genetic reagent (*D. melanogaster*) | Rbcn-3A-RNAi | Vienna *Drosophila* Resource Center | V108547 | |
| Genetic reagent (*D. melanogaster*) | Rab7-RNAi | Vienna *Drosophila* Resource Center | V40337 | |
| Genetic reagent (*D. melanogaster*) | UAS-Hrs-RNAi | Vienna *Drosophila* Resource Center | V20933 | |
| Genetic reagent (*D. melanogaster*) | UAS-mega/pickel-RNAi | Vienna *Drosophila* Resource Center | V36306 | |
| Genetic reagent (*D. melanogaster*) | hhGal4 | Bloomington *Drosophila* Stock Center | BL67046 | |
| Genetic reagent (*D. melanogaster*) | tubGal80$^{ts}$ | Bloomington *Drosophila* Stock Center | BL7108 | |
| Genetic reagent (*D. melanogaster*) | UAS-p35 | Bloomington *Drosophila* Stock Center | BL5072 | |
| Genetic reagent (*D. melanogaster*) | enGal4 | Bloomington *Drosophila* Stock Center | BL30564 | |
| Genetic reagent (*D. melanogaster*) | Lac::GFP | Gift from Christian Klämbt, University of Münster, Germany | | Endogenously GFP-tagged *lac* allele |
| Genetic reagent (*D. melanogaster*) | ATPα::GFP | Bloomington *Drosophila* Stock Center | BL6834 | |
| Genetic reagent (*D. melanogaster*) | NrxIV::GFP | *Edenfeld et al., 2006* | | |

*Continued on next page*

*Continued*

| Reagent type (species) or resource | Designation | Source or reference | Identifiers | Additional information |
|---|---|---|---|---|
| Genetic reagent (*D. melanogaster*) | Nrg::GFP | Bloomington *Drosophila* Stock Center | BL6844 | |
| Genetic reagent (*D. melanogaster*) | FasIII::GFP | Bloomington *Drosophila* Stock Center | BL59809 | |
| Genetic reagent (*D. melanogaster*) | Vari::Dendra | *Babatz et al., 2018* | | Endogenously Dendra-tagged *vari* allele |
| Genetic reagent (*D. melanogaster*) | Mega::YFP | Gift from Reinhard Schuh, Max Planck Institute for Biophysical Chemistry, Göttingen, Germany | | Endogenously YFP-tagged *mega* allele |
| Genetic reagent (*D. melanogaster*) | Vps35::RFP | Bloomington *Drosophila* Stock Center | BL66527 | |
| Genetic reagent (*D. melanogaster*) | Vps26-EGFP | Bloomington *Drosophila* Stock Center | BL67153 | |
| Genetic reagent (*D. melanogaster*) | UAS-Shi$^{K44}$ | Bloomington *Drosophila* Stock Center | BL5811 | |
| Genetic reagent (*D. melanogaster*) | Ubx-Flp | Bloomington *Drosophila* Stock Center | BL42730 | |
| Genetic reagent (*D. melanogaster*) | hsFlp | Bloomington *Drosophila* Stock Center | BL8862 | |
| Genetic reagent (*D. melanogaster*) | hsFlp FRT19A ubi-nls-RFP | Bloomington *Drosophila* Stock Center | BL31418 | |
| Genetic reagent (*D. melanogaster*) | FRT40A 2xGFP | Bloomington *Drosophila* Stock Center | BL5189 | |
| Genetic reagent (*D. melanogaster*) | FRT42D 2xGFP | Bloomington *Drosophila* Stock Center | BL5625 | |
| Genetic reagent (*D. melanogaster*) | FRT42D ubi-nls-RFP | Bloomington *Drosophila* Stock Center | BL35496 | |
| Genetic reagent (*D. melanogaster*) | FRT82B ubi-nls-RFP | Bloomington *Drosophila* Stock Center | BL30555 | |
| Genetic reagent (*D. melanogaster*) | FRT42D Vps35$^{MH20}$ | Bloomington *Drosophila* Stock Center | BL67202 | |
| Genetic reagent (*D. melanogaster*) | Vps26$^B$ FRT19A | Bloomington *Drosophila* Stock Center | BL57140 | |
| Genetic reagent (*D. melanogaster*) | Hrs$^{D28}$ Stam$^{2L2896}$ FRT40A | *Tognon et al., 2014* | BL56816 | |
| Genetic reagent (*D. melanogaster*) | Dmon1$^{Mut4}$ FRT40A | *Yousefian et al., 2013* | | |
| Genetic reagent (*D. melanogaster*) | Snx27$^{25}$ FRT19A | *Strutt et al., 2019* | | |
| Genetic reagent (*D. melanogaster*) | FRT42 Fam21$^{KO}$ | *Strutt et al., 2019* | | |
| Genetic reagent (*D. melanogaster*) | Snx1$^{Δ2}$ FRT40A | *Zhang et al., 2011* | | |
| Genetic reagent (*D. melanogaster*) | Snx6$^1$ FRT40A | *Zhang et al., 2011* | | |
| Genetic reagent (*D. melanogaster*) | FRT82B Snx3$^{EY05688}$ | *Harterink et al., 2011* | | |

*Continued on next page*

*Continued*

| Reagent type (species) or resource | Designation | Source or reference | Identifiers | Additional information |
|---|---|---|---|---|
| Genetic reagent (*D. melanogaster*) | *Rab7*$^{GAL4-KO}$ | *Chan et al., 2011* | | |
| Antibody | anti-ATPα (mouse monoclonal) | DSHB | a5 | IF (1:50) |
| Antibody | anti-Shrub (rabbit polyclonal) | *Bäumers et al., 2019* | | IF (1:100) |
| Antibody | anti-Mega (mouse monoclonal) | *Jaspers et al., 2012* | | IF (1:50) |
| antibody | anti-E-cad (rat monoclonal) | DSHB | 5D3 | IF (1:50) |
| Antibody | anti-Rab7 (mouse monoclonal) | DSHB | Rab7 | IF (1:100) |
| Antibody | anti-Dlg (mouse monoclonal) | DSHB | 4F3 | IF (1:500) |
| antibody | anti-Cora (mouse monoclonal) | DSHB | C566.9 | IF (1:100) |
| Antibody | anti-Vps26 (guinea pig polyclonal) | *Wang et al., 2014* | | IF (1:1500) |
| Antibody | anti-Chc (rat polyclonal) | *Wingen et al., 2009* | | IF (1:50) |
| Antibody | anti-Golgin84 (mouse monoclonal) | DSHB | 12–1 | IF (1:100) |
| Antibody | anti-Lgl (guinea pig polyclonal) | *Shahab et al., 2015* | | IF (1:500) |
| Antibody | anti-Cont (guinea pig polyclonal) | *Faivre-Sarrailh et al., 2004* | | IF (1:2000) |
| Antibody | anti-Notch (mouse monoclonal) | DSHB | C458.2H | IF (1:100) |
| antibody | anti-Hrs (guinea pig polyclonal) | *Lloyd et al., 2002* | | IF (1:500) |
| Antibody | anti-Ubiquitin (FK2) (mouse monoclonal) | Enzo Life Sciences | BML-PW8810 | IF (1:100) |
| Antibody | anti-Crb (rat polyclonal) | *Richard et al., 2006* | rat anti-Crb 2.8 | IF (1:500) |
| Antibody | anti-HA (rabbit monoclonal) | Cell Signalling Technology | C29F4 | IF (1:1500) |
| Antibody | anti-Rab5 (rabbit polyclonal) | Abcam | ab31261 | IF (1:250) |
| Antibody | anti-Fmi (mouse monoclonal) | DSHB | #74 | IF (1:10) |
| Sequence-based reagent | fw-NotI-ATG-HA-3xGly-Mega | This paper | PCR primer | See method section for sequence |
| Sequence-based reagent | rev-Mega-XhoI | This paper | PCR primer | See method section for sequence |
| Other | Hoechst 33258 | Sigma Aldrich/ Merck | B2883 | IF (1:10.000), DNA/Nucleus staining |

## *Drosophila* stocks and genetics

A complete list of all stocks used in this study is found in the key resources table above. Flies were raised on standard cornmeal/molasses/yeast diet and kept at room temperature. Crossings were raised on 25°C, except for experiments containing the temperature sensitive *tubGal80ts* (*McGuire et al., 2004*). Those flies were kept at 18°C (permissive temperature) to inhibit Gal4/UAS-

mediated expression (*Brand and Perrimon, 1993*) and shifted to 29°C (restrictive) for specific time spans to activate UAS-based expression. Flp/FRT system (*Xu and Rubin, 1993*) induced clones were either generated by expression of Ubx-FLP or using hs-FLP with an 70 min heatshock in the first instar larval stage (24–48 hr after egg laying).

## Generation of transgenic flies

For generation of *UAS* and *tubP* expressed HA-tagged Mega constructs, the clone LD14222 from *Drosophila* Genomics Resource Center (DGRC) was used as cDNA source. The HA-tag followed by a 3xGlycin linker was fused to the N-terminus of the Mega open reading frame by PCR using the following primers: fw-NotI-ATG-HA-3xGly-Mega GgcggccgcatgTACCCATACGAcGTTCCAGAcTACGC TggcggggcCGCGAACTTAACAAGCAGCAG rev-Mega-XhoI ctgcactcgagTTATATGTAGCCC TGCAGGC.

The generated PCR fragment was restriction digested with NotI and XhoI enzymes (New England Biolabs) and subsequently cloned into pUAST-attB and pTUB vectors. The tubulinP-based plasmid derived from a pCaSpeR4-tubulin-QF#7 vector (addgene) with SV40 polyA 3' UTR. The UAS-HA-Mega$^{K2R}$ construct was designed analogous to UAS-HA-Mega with all intracellular lysines exchanged to arginines. The cDNA for this construct was synthesized by BaseClear B.V. Generation of transgenic flies was performed by attB/attP specific genomic integration into the landing sites 51C (for 2nd chromosome) and 86Fb (for 3rd chromosome) (*Bischof et al., 2007*). Injection of embryos was either performed in house or by BestGene Inc (CA 91709, U.S.A).

## Immunohistochemistry (IHC)

A complete list of all antibodies used in this study is found in the key resources table above. Late L3 larvae or pupae were dissected in phosphate-buffered saline (PBS, pH 7.4) on ice and immediately fixed for 30 min in 4% paraformaldehyde in PBS. Following three 10 min wash steps in PBT (0.3% Triton X-100 in PBS), tissue was blocked with 5% normal goat serum (NGS, Jackson ImmunoResearch) in PBT and subsequently incubated with primary antibodies in 5% NGS in PBT for 2 hr. After three 15 min wash steps with PBT, discs were incubated with fluorochrome-conjugated antibodies (Alexa-488, −568, −647 from Thermo Fisher Scientific) for 2 hr in 5% NGS in PBT. For nuclear staining, Hoechst 33258 (Sigma Aldrich) was used at a concentration of 1:10,000. Discs or pupae were mounted in Vectashield (Vector Laboratories) and imaged with a Zeiss AxioImager Z1 wide field microscope equipped with a Zeiss Apotome. For SJ length measurements, ROIs were manually assigned with Fiji and the total length of electron-dense SJ within ROIs was measured to yield a SJ length/ROI length ratio. Statistical analysis was performed with GraphPad Prism 7.0 d.

## Live imaging

Late L3 instar larvae were dissected in PBS and immediately mounted on coverslips in Schneider's *Drosophila* medium (Pan Biotech). Double-sided tape containing a punchhole was used both as a spacer to avoid tissue damage during mounting and as a short-term imaging chamber that restricted wing disc movement. Imaging was performed with a Zeiss LSM 880 laser scanning microscope equipped with an Airyscan detector. For time series, frames were acquired in 1–4 s intervals and line switching was used for dual channel acquisition.

## Fluorescence recovery after photobleaching (FRAP)

Wing discs were mounted as described above (see Live Imaging). Photobleaching and imaging were performed on a Zeiss LSM 880 confocal microscope using a gallium arsenide phosphide (GaAsP) detector and 40x objective with a numeric aperture of 1.2. A 15x digital zoom was applied to yield a detection area of 14.17 × 14.17 micron, equivalent to 512 × 512 pixels. In the focal plane of the junctions, a bicellular SJ ROI (0.83 × 0.83 microns) was bleached with 100% excitation laser intensity and 10 repeats. The laser dwell time for each pixel was 4.1 µs. After bleaching, the tissue was imaged in 2 min intervals for 30 min and any potential drifting (x-, y-, z-axis) was corrected manually. Images were merged into a time series with Fiji and the mean gray values within the ROI were plotted against time/min (*Figure 1J*). Statistical analysis was performed with Microsoft Excel and GraphPad Prism 7.0 d.

## Vesicle tracking

The Fiji plugin TrackMate (*Tinevez et al., 2017*) was used on bleach corrected (exponential fit) time series obtained by live imaging to analyze movement of vesicles positive for either Vps26-EGFP or Vps35::RFP. A Dog detector with estimated blob diameter of 2 micron and threshold 1 were used as a fixed setup. Tracks were analyzed by the simple LAP tracker with following parameters: Linking max distance: 2. Gap closing distance: 2. Gap closing max frame gap: 20. Mean velocity was used as the parameter for comparing vesicles within wildtype and Shrub-depleted tissue. Statistical analysis was performed with GraphPad Prism 7.0 d.

## Transmission electron microscopy (TEM)

Wing discs were dissected in 0.1M phosphate buffer (0.1M $PO_4$) on ice and immediately fixed with 2.5% glutaraldehyde in 0.1M $PO_4$ for 1 hr. Following five 5 min washsteps with 0.1M $PO_4$, tissue was postfixed with osmium tetroxide (2% in 0.1M $PO_4$) for 1 hr on ice. After three 5 min washsteps with 0.1M $PO_4$ and further washsteps with dd$H_2O$, the specimens were gently dehydrated with ethanol in concentrations ranging from 50 to 100%. Acetone was used as an intermediate solvent to support Epoxy Embedding Medium (Epon, Sigma Aldrich) infiltration into the tissue. Specimens were stored over night in 100% Epon and subsequently embedded and sectioned. Semi-thin sections were stained with Richardson blue. Ultra-thin sections were contrasted with 2% uranyl acetate and lead citrate prior to imaging. Sections were analyzed with an EM 902 (Zeiss) microscope at 80 KV.

## Fluorescence in situ hybridization (FISH)

For generation of Mega RNA antisense probe, the Mega/Pickel cDNA containing LD14222 clone from DGRC was linearized by EcoRI digestion following T7 polymerase-dependent in vitro transcription and digoxigenin (DIG) labeling. RNA probes were purified with a NucleoSpin Gel and PCR clean-up kit before use in the hybridization reaction. For FISH, late L3 larvae were dissected in PBS on ice and fixed for 20 min in 3.7% formaldehyde in PBS. Specimens were handled according to standard protocols. Briefly, larval tissue was incubated with RNA probes in formamide-based hybridization buffer at 65°C over night. To detect probes, HRP-conjugated anti-DIG antibodies (Perkin Elmer) were used, followed by a tyramide signal amplification reaction (TSA plus Cyanine 3 Fluorescence Kit, Perkin Elmer). Wing discs were mounted in Vectashield (Vector Laboratories) and imaged with a Zeiss AxioImager Z1 wide field microscope equipped with a Zeiss Apotome.

## Dye penetration assay

Late L3 larvae were dissected in PBS and briefly rinsed (30 s) in PBS containing 10 kDa Texas-dsRed labeled Dextran (Molecular Probes D-1828) in a concentration of 1 mg/ml. Following two quick wash steps in PBS, imaginal discs were dissected from the carcasses and immediately mounted in Vectashield (Vector Laboratories) on microscope slides. To avoid tissue damage due to compression, double-sided tape was used as a spacer between slide and cover slip. Images were acquired on a Zeiss AxioImager Z1 wide field microscope immediately after mounting (within a timeframe of 5 min).

## Colocalization analysis

For measuring colocalization of Mega::YFP with either Vps26 or Chc, dual channel images of single fluorescent planes (either lateral views or basal planes) were adjusted for brightness in FiJi. Colocalization was manually determined for each individual Mega::YFP vesicle using the FiJi cell counter plugin.

## Acknowledgements

We thank Jessica Hausmann, Stefan Kölzer and Sylvia Tannebaum for excellent technical support. We would like to thank Reinhard Schuh, Fen-Biao Gao, Christian Klämbt, David Strutt, Peter J Cullen, Xinhua Lin, Konrad Basler, Hugo Bellen, Matthias Behr, Andreas Wodarz, Manzoor A Bhat and Elisabeth Knust for kindly sharing fly stocks and reagents. We thank the Center for Advanced Imaging (CAi) at the Heinrich-Heine-University for technical support with microscopy and access to imaging equipment. The Bloomington *Drosophila* Stock Center, Vienna *Drosophila* Resource Center,

*Drosophila* Genomics Resource Center and the Developmental Studies Hybridoma Bank supplied fly-stocks, plasmid DNA, and antibodies.

## Additional information

### Funding

| Funder | Grant reference number | Author |
| --- | --- | --- |
| Deutsche Forschungsge-meinschaft | Sachbeihilfe KL 1028/11-1 | Hendrik Pannen<br>Tim Rapp<br>Thomas Klein |

The funders had no role in study design, data collection and interpretation, or the decision to submit the work for publication.

### Author contributions

Hendrik Pannen, Conceptualization, Data curation, Software, Funding acquisition, Investigation, Visualization, Methodology, Writing - original draft, Writing - review and editing; Tim Rapp, Data curation, Software, Investigation, Visualization, Methodology; Thomas Klein, Conceptualization, Resources, Supervision, Funding acquisition, Methodology, Writing - original draft, Project administration, Writing - review and editing

### Author ORCIDs

Hendrik Pannen (iD) https://orcid.org/0000-0003-0641-0804
Thomas Klein (iD) https://orcid.org/0000-0002-2719-9617

### Decision letter and Author response

Decision letter https://doi.org/10.7554/eLife.61866.sa1
Author response https://doi.org/10.7554/eLife.61866.sa2

## Additional files

### Supplementary files

• Transparent reporting form

### Data availability

All data generated or analysed during this study are included in the manuscript and supporting files.

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
