## [Decision Letter]

**Acceptance summary:**

By using temporal and spatial resolution of knockdown the authors studied sub-cellular changes and protein trafficking events induced by loss of ESCRT-III in *Drosophila* imaginal discs prior to the massive disruption of cellular architecture and neoplastic overgrowth. The data unveil a novel function of the ESCRT III core machinery in localising Retromer positive vesicles apically. They further uncover an ESCRT III- and Retromer-dependent transcytosis-like mechanism in basal to apical transport of the Claudin Megatracheae to the SJ, a prerequisite for the maintenance of junctional and tissue structure.

**Decision letter after peer review:**

Thank you for submitting your article "The ESCRT Machinery regulates Retromer dependent Transcytosis of Septate Junction Components in *Drosophila*" for consideration by *eLife*. Your article has been reviewed by two peer reviewers, one of whom is a member of our Board of Reviewing Editors, and the evaluation has been overseen by Utpal Banerjee as the Senior Editor. The following individual involved in review of your submission has agreed to reveal their identity: Raghu Padinjat (Reviewer #2).

The reviewers have discussed the reviews with one another and the Reviewing Editor has drafted this decision to help you prepare a revised submission.

Summary:

The manuscript presented by Pannen et al. addresses the question on the role of trafficking in epithelia (here: the wing disc of *Drosophila* larvae). In particular the authors revisit the consequence of loss of the ESCRT-complex on the integrity of septate junctions (SJ), which may potentially have an impact on the induction of neoplastic overgrowth. The major findings are: i) Loss of Shrub, a major component of ESCRT, results in defects in septate junction formation and function by preventing the proper apical delivery of Megatrachea (Mega), a SJ component. ii) Loss of Shrub/ESCRT affects the proper apical localization of Retromer. iii) Improper localization of Retromer, in turn, affects transcytosis of Mega and other SJ components from the basal to the apical side. These are novel data, interesting for people working on trafficking in epithelia and tumour formation.

Essential revisions:

Overall this is an interesting, excellent and substantial body of work. The major conclusions put forward are justified by the data. Experiments are well done and the results presented very clearly. There are a few points that the authors may want to consider:

1) The authors themselves point out that many elements of the pathway described here have already been presented in the context of polarized cells in the *Drosophila* embryo. They highlight the importance of their work in the wing disc as the demonstration of these findings in the epithelium of a mitotic cell that is actively dividing. Bearing this in mind, could they present the effect of retromer depletion on wing disc morphology? Will depletion of retromer components result in the neoplastic phenotype presented in Figure 1B, C. This would be a small but important addition.

2) It is proposed that Mega RNA is enriched near the basal (lateral) membrane, transcytosed and delivered to the apical domain. The authors have pointed out that it is unclear why this complex route of protein trafficking is used. It would be good to discuss why this is the case, i.e. why does it have to be translated near the basal membrane. Does it mean that some other basal membrane component is needed to associate with Mega prior to its transport to the apical domain?

3) Figure 3G: I suggest to quantify the overlap between Mega-YFP and Vps26 to make the conclusion more robust.

Points that could contribute to clarification (Please address if this is feasible. At least address the issues in the revised draft)

1) Figure 1H/subsection “ESCRT knockdown specifically affects SJ integrity”: This figure is not clear, I guess H and H' show the same part of the disc? And what does the inset show? Part of the same image? What does the sentence mean "This basal fraction of Mega was strongly labelled with the endosomal marker Rab7.… "? Does "strongly" mean co-localisation? Could this quantified?

2) Subsection “ESCRT knockdown specifically affects SJ integrity”: the role of Mega with respect to localising other SJ components is deduced from its role in embryos. What about the consequences of loss of Mega in discs?

3) Figure 1I: is the mis-localised Lac-GFP really localised on the lateral membrane? This is only obvious on sites indicated by the arrowheads. What are the blobs seen in other cases?

4) Figure 1J: what are the error bars in this figure?

5) Subsection “The Retromer CSC regulates membrane levels of SJ core components”: the authors show that loss of retromer components differentially affects components of the SJs. This raises two questions: i) how does loss of shrub affect other components, e. g. Dlg or Cora? ii) Does the reduction of Mega upon loss of retromer affect the structure of the SJ?

6) Figure 4E: I suggest to quantify the overlap between Mega-YFP and Chc to make the conclusion more robust, in particular to demonstrate that the overlap is predominantly on the basal side.

7) The authors write that transcytosis "appears to serve as a universal mechanism. ". I find this conclusion somewhat premature. Tiklova et al., 2010 only checked for Cora as a SJ marker.

8) In the manuscript here, Cora seems to be unaffected upon loss of Vps35 (Figure 2F). So different SJ components behave differently upon blocking endocytosis in embryonic and larval epithelia. It would also be interesting to get to know the opinions of the authors with respect to the different behaviour of SJ components in their experiments.

9) Along the same line: what is the authors' opinion on the function of the retromer CSC (Vps35, Vps26), which seems to act independent of other retromer components (e.g. sorting nexins, as shown in Figure 2—figure supplement 5)?

---

## [Author Response]

Essential revisions:Overall this is an interesting, excellent and substantial body of work. The major conclusions put forward are justified by the data. Experiments are well done and the results presented very clearly. There are a few points that the authors may want to consider:1) The authors themselves point out that many elements of the pathway described here have already been presented in the context of polarized cells in the *Drosophila* embryo. They highlight the importance of their work in the wing disc as the demonstration of these findings in the epithelium of a mitotic cell that is actively dividing. Bearing this in mind, could they present the effect of retromer depletion on wing disc morphology? Will depletion of retromer components result in the neoplastic phenotype presented in Figure 1B, C. This would be a small but important addition.

Depletion of the Retromer component Vps26 does not induce neoplastic transformation in wing imaginal disc tissue. We have added a semi thin section of a Vps26-RNAi expressing wing disc to Figure 2—figure supplement 4, showing the intact epithelial monolayer organization. These are the discs we used to quantify SJ density, which revealed a significant reduction in the amount of electron dense septa found in the posterior Vps26 depleted compartment. Thus, although having defects in SJ integrity, Retromer mutant / depleted tissue does not lose apicobasal cell polarity (also note the wildtype E-cad and Dlg levels within Vps35 mutant clones; Figure 2). We believe that the major driver of neoplasia in ESCRT deficient wing discs is the high amount of simultaneously misregulated signaling pathways (such as ectopic Notch activation, overview in (Vaccari and Bilder, 2009)). In contrast, Vps26 depletion does not lead to ectopic Notch signaling activation (Gomez-Lamarca et al., 2015), Thus, an important difference between Retromer and ESCRT loss / depletion is the impact on cellular signaling.

2) It is proposed that Mega RNA is enriched near the basal (lateral) membrane, transcytosed and delivered to the apical domain. The authors have pointed out that it is unclear why this complex route of protein trafficking is used. It would be good to discuss why this is the case, i.e. why does it have to be translated near the basal membrane. Does it mean that some other basal membrane component is needed to associate with Mega prior to its transport to the apical domain?

At this stage we can only speculate about the reason for Mega being translated basally and initially transported to the basodistal membrane. Currently we do not know whether this route is specific to Mega or shared with other SJ core components. It is possible that the milieu at the basodistal membrane facilitates SJ complex formation, which could require SJ components to be translated in proximity. Alternatively, basal subcellular localization of Mega transcripts could reflect early / transcriptional spatial separation of apical (such as E-cad) and basolateral (such as SJ components) cargos. This would mean that the complex transcytosis route is an adaptation for tissues in which apical and basolateral cargos reside in close proximity at the apical pole of the cells (as is the case in imaginal discs). It would be interesting to test the transcript localizations of other SJ components and compare those to apical cargos such as Notch, Crumbs and E-cad. At this stage, these hypotheses remain highly speculative but we added these to the Discussion for the revised manuscript.

3) Figure 3G: I suggest to quantify the overlap between Mega-YFP and Vps26 to make the conclusion more robust.

We have quantified the overlap of Mega::YFP vesicles with the Retromer component Vps26, which revealed a 71,9% Vps26 positivity rate for Mega::YFP vesicles throughout wing imaginal disc cells. Additionally, we included a live-imaging video showing colocalization and co-movement of Mega::YFP with Vps35::RFP (Animation 1).

Points that could contribute to clarification (Please address if this is feasible. At least address the issues in the revised draft)1) Figure 1H/subsection “ESCRT knockdown specifically affects SJ integrity”: This figure is not clear, I guess H and H' show the same part of the disc? And what does the inset show? Part of the same image? What does the sentence mean "This basal fraction of Mega was strongly labelled with the endosomal marker Rab7.…"? Does "strongly" mean co-localisation? Could this quantified?

Indeed, Figure 1H and H´ show two channels (Mega and Rab) of the same basal part of the wing disc tissue. The inset (both channels merged, colocalization in white) is a magnification of some of these basal endosomes that show large overlap between Mega and Rab7. We agree that this was not very evident for the reader, so we revised this part of the figure by outlining the origin of the inset. The strong colocalization of Mega / HA-Mega with Rab7 on basal aberrant endosomal compartments induced by Shrub depletion is also shown in Figure 3—figure supplement 2.

2) Subsection “ESCRT knockdown specifically affects SJ integrity”: the role of Mega with respect to localising other SJ components is deduced from its role in embryos. What about the consequences of loss of Mega in discs?

We now have included a figure that shows slight lateral spreading of Lac::GFP upon Mega depletion in wing imaginal discs (Figure 1—figure supplement 2). This suggests that similar to SJ in the embryo, the imaginal disc SJ complex (partially) loses its integrity and confined localization upon loss of individual SJ components. Since this phenotype is much weaker compared to Lac::GFP lateral spreading upon Shrub depletion (Figure 1I), the data also indicate that loss of Mega alone is insufficient to strongly destabilize Lac::GFP within the junction. Thus, the more severe Lac::GFP phenotype observed upon Shrub depletion suggests that besides Mega, apical levels of more SJ core components might be affected in ESCRT depleted tissue. We added another figure in which we confirmed this by visualizing the SJ core component ATPα in Shrub depleted wing disc tissue, revealing a strong reduction at the SJ level (Figure 1—figure supplement 3), similar to the phenotype observed for Mega (Figure 1F).

3) Figure 1I: is the mis-localised Lac-GFP really localised on the lateral membrane? This is only obvious on sites indicated by the arrowheads. What are the blobs seen in other cases?

The lateral spreading of Lac::GFP is particularly evident at the level of the nuclei. The plasma membrane at this level is devoid of Lac::GFP in the anterior control compartment but enriched with Lac::GFP in the Shrub depleted compartment. The phenotype might be more apparent in the upper half of the image, which shows parts of the epithelium that do not represent a sagittal section. We have added further arrows to highlight this part of the image, which might help in understanding this phenotype easier. Also note how this lateral membrane fraction (in the upper part of the image) sharply appears at the a/p compartment boundary. The Lac::GFP blobs are most likely endosomal compartments, although we did not stain for the endosomal marker Rab7 in this particular experiment. Note that most TMPs and / or membrane anchored proteins that we analyzed show endosomal accumulation to varying extents upon ESCRT loss / depletion.

4) Figure 1J: what are the error bars in this figure?

The error bars indicate standard deviation from the mean fluorescence signal intensity at each individual time point. We have included this in the figure legend.

5) Subsection “The Retromer CSC regulates membrane levels of SJ core components”: the authors show that loss of retromer components differentially affects components of the SJs. This raises two questions: i) how does loss of shrub affect other components, e. g. Dlg or Cora?

The error bars indicate standard deviation from the mean fluorescence signal intensity at each individual time point. We have included this in the figure legend.

ii) Does the reduction of Mega upon loss of retromer affect the structure of the SJ?

Loss of Retromer does not only affect Mega junctional levels, but also those of other SJ core components (see Figure 2). The effect of Retromer depletion on SJ density is displayed in Figure 2—figure supplement 4. The reduction of SJ density by about 50% also makes Retromer depleted wing disc tissue leaky, indicated by defective epithelial barrier function (Figure 1—figure supplement 1).

6) Figure 4E: I suggest to quantify the overlap between Mega-YFP and Chc to make the conclusion more robust, in particular to demonstrate that the overlap is predominantly on the basal side.

We have quantified the amount of Mega::YFP vesicles that overlap with Chc (colocalization) and included this data in the revised draft. Quantification revealed a Chc positive rate of 55,2% for Mega::YFP vesicles throughout the entire cell, while the rate was slightly increased to 63,01% in basal planes of the epithelium. This is consistent with Mega::YFP undergoing clathrin dependent endocytosis at the basodistal membrane and being trafficking in clathrin coated vesicles along the apico-basal axis in wing imaginal discs.

7) The authors write that transcytosis "appears to serve as a universal mechanism. ". I find this conclusion somewhat premature. Tiklova et al., 2010 only checked for Cora as a SJ marker.

We agree with the reviewer that this statement is premature. We have revised the section, now stating that transcytosis is likely required for both initial formation and maintenance (e.g. in imaginal discs) of SJ in *Drosophila*. However, besides Cora, Tiklova and colleagues also showed a similar effect of clathrin loss on Melanotransferrin / Mtf localization (lateral membrane spreading) (Tiklova et al., 2010). Although not being a TMP, the GPI-anchor of Mtf suggests a tight membrane association of this SJ component, which sets it apart from cytoplasmic SJ associated proteins such as Dlg and Cora (Tiklova et al., 2010). Unfortunately, the Mtf antibody did not work in our hands on imaginal disc tissue to test whether its SJ delivery in wing discs requires ESCRT / Retromer functions.

8) In the manuscript here, Cora seems to be unaffected upon loss of Vps35 (Figure 2F). So different SJ components behave differently upon blocking endocytosis in embryonic and larval epithelia. It would also be interesting to get to know the opinions of the authors with respect to the different behaviour of SJ components in their experiments.

It is indeed an interesting observation that concentration of Cora during SJ formation in the embryo requires clathrin / endocytosis (Tiklova et al., 2010) while junctional Cora levels in imaginal discs are independent of Retromer (Figure 2F). If we assume a similar pathway relying on endocytosis and potentially Retromer dependent junctional delivery of SJ TMP components in larval tissues and the embryo, this could mean that clathrin deficient embryos might fail to initially form septa at all (due to the lack of SJ TMP complex trafficking). The potential lack of septa might prevent cytoplasmic scaffolding components such as Cora from concentrating at the SJ membrane region. In tissues with stably established polarity and junctional integrity (imaginal discs), even a small amount of SJ complexes / septa residing at the junctional region (e.g. by introduction of Vps35/Vps26 null mutant clones) might be sufficient to serve as a positional cue for scaffolding components. It is therefore possible that Retromer function in the embryo is even more essential to SJ formation than it is for maintenance in proliferating tissues. This of course is hypothetical and demands investigation. It is important to note that FRAP kinetics of Cora in the embryo were similar to FRAP rates of SJ TMP core components (Oshima and Fehon, 2011). Thus, Cora may be part of the core SJ complex in embryos and consequently dependent on the TMP components for its localization. Our data in wing discs however support the idea that Cora might not traffic in complex with the TMP SJ components in wing imaginal discs but is rather independently targeted to the SJ. It would be interesting to test whether Cora FRAP kinetics in the wing disc lean more towards the TMP fraction or if they resemble the quick recovery kinetics of Dlg (Oshima and Fehon, 2011). This could help unravel a potential difference in core complex composition between embryonic and larval SJ.

9) Along the same line: what is the authors' opinion on the function of the retromer CSC (Vps35, Vps26), which seems to act independent of other retromer components (e.g. sorting nexins, as shown in Figure 2—figure supplement 5)?

The data we present in this manuscript suggest that the CSC is required to export Mega (and likely other TMP SJ core components) from the endosomal system to target the proteins to the junctional membrane. Since the CSC itself does not possess membrane deforming activities, we can only speculate on how Mega is sorted out of the maturing endosome. It is possible that previously uncharacterized CSC co-factors may aid Retromer during formation of Mega containing transport carriers / tubules. Alternatively, CSC interaction with Mega / SJ components within maturing endosomes may save these proteins from the degradative ILV route without actually exporting them from the endosome, although it is unclear how they could reach the SJ directly from maturing endosomes (without formation of a specific CSC/SJ transport carrier). Thus, we favor the model of a classical Retromer export route (formation of a CSC dependent carrier vesicle / tubule from maturing endosomes), with the detailed mechanism of carrier generation being obscure. We have discussed this briefly in the revised manuscript.

References:

Gomez-Lamarca, M.J., L.A. Snowdon, E. Seib, T. Klein, and S.J. Bray. 2015. Rme-8 depletion perturbs Notch recycling and predisposes to pathogenic signaling. J Cell Biol. 210:303-318.Vaccari, T., and D. Bilder. 2009. At the crossroads of polarity, proliferation and apoptosis: the use of *Drosophila* to unravel the multifaceted role of endocytosis in tumor suppression. Mol Oncol. 3:354-365.